# Using machine learning to predict persecutory beliefs based on aetiological models of delusions identified in a systematic literature search

**Saskia Denecke** ✉, **Felix Strakeljahn** , **Antonia Bott** & **Tania M. Lincoln**

Aetiological models of delusions propose a broad range of predictors. The extent to which these predictors explain variance in persecutory beliefs across the continuum requires systematic investigation. As part of a previous review, 51 aetiological models of delusions were identified in a systematic literature search using PubMed, Web of Science, and Science Direct databases. Omitting repetitions, 66 unique postulated predictors of delusions and persecutory delusions were extracted from these models, of which 55 met our inclusion criteria and were assessed in a cross-sectional online sample stratified by delusion severity ($N = 336$) using self-report and behavioural measures. Utilising machine learning (i.e., random forests with nested cross-validation), we investigated the extent to which the model-based predictors explain self-reported persecutory beliefs, identified the most relevant predictors, and investigated their specificity in explaining persecutory beliefs as opposed to delusional beliefs or psychopathological symptoms in general. The machine learning model explained 31% of the variance in persecutory beliefs, 47% of delusions in general, and 77% of general psychopathology. The ten predictors with the most influence on predicting persecutory beliefs included negative beliefs about mistrust, cognitive fusion, ostracism, threat anticipation, generalised negative other beliefs, trust, aberrant salience, hallucinations, stress, and emotion regulation difficulties. The limited explanatory power of the proposed predictors raises questions about the validity of existing models and suggests that crucial predictors specific to persecutory delusions may be missing. Our findings highlight the importance of investigating, refining, and cross-validating theoretical aetiological models to improve our understanding of the aetiology of delusions.

Delusions are fixed beliefs that are held despite contrasting evidence[1]. They are one of the core symptoms of psychosis and cause significant distress[2,3]. Persecutory delusions, the belief that other individuals or groups intend to harm or plot against oneself[1], are the most common[4]. Other common delusions are delusions of reference (i.e., the belief that innocuous events are signs directed at oneself[1]), which are related to but differentiable from persecutory delusions. Mild forms of persecutory beliefs are prevalent in the general population[5,6] and are commonly thought to lie on a continuum ranging from mild ideation to manifest delusions[7]. Delusions remain challenging to treat. In meta-analyses, psychological approaches, such as cognitive behavioural therapy, have demonstrated small to medium effects in treating delusions that are not maintained over longer follow-up

periods[8,9]. To refine and develop targeted treatments, it is necessary to understand the factors that contribute to delusion formation and maintenance.

Numerous theoretical models aim to explain the aetiology of delusions in general and persecutory delusions specifically. As summarised in a recent review of 53 models of delusions[10], these take on different perspectives and have emphasised different mechanisms while agreeing on a shared key mechanism. Specifically, the models propose that delusions are formed to explain an experienced anomalous event. Postulated predictors of delusions range from cognitive aspects (e.g., generalised beliefs about the self and other people and reasoning biases) over social factors (e.g., social exclusion and deficits in social cognition) and Bayesian inference indices (e.g., weighing of

Clinical Psychology and Psychotherapy, University of Hamburg, Hamburg, Germany. ✉e-mail: Saskia.denecke@outlook.com

prior beliefs and new information) to neurobiological aberrations (e.g., neurotransmitter dysregulation) and factors related to associative learning (e.g., attentional biases and conditioning). Thus, an extensive range of predictors proposed to contribute to the aetiology of delusions has accumulated over the years, giving the impression that the field has advanced in understanding the aetiology of delusions.

However, despite extensive research on individual predictors, aetiological models as a whole are infrequently tested empirically. Although some studies have used a selection of predictors drawn from theoretical models[11,12], the predictive value of all predictors derived from a theoretical model, let alone the complete set of predictors from all models, has not been systematically examined. Therefore, we do not know whether and to what extent persecutory beliefs specifically and delusional beliefs in general can be explained by combining different models and predictors. It would probably be unreasonable to expect that the models explain 100% of the variance due to various influences, such as measurement error and the complexity of psychological phenomena. Nonetheless, investigating the combined explained variance would provide an estimate of how well current models cover factors relevant to persecutory beliefs. We further lack an understanding of which factors are most relevant and which might have no unique predictive value and could be omitted from aetiological models. To evaluate and refine the models and identify potentially relevant treatment targets, it is necessary to examine the factors proposed by existing theoretical models systematically.

Therefore, to follow up on the theoretical summary in Denecke et al.[10], this study aims to (Q1) provide an estimate of the variance in persecutory beliefs currently accounted for by the diverse aetiological models of delusions, (Q2) pinpoint predictors bearing the highest relevance for explaining persecutory beliefs, (Q3a) investigate how well individual scores of persecutory beliefs can be explained, and (Q3b) examine whether more parsimonious models can predict persecutory beliefs just as well as the complete model. Further, we aim to (Q4) explore whether the proposed risk factors are specific to persecutory beliefs compared to delusions in general or to a broader range of psychopathological symptoms. To reach these aims, we utilised machine learning (ML) models (i.e., random forests) to examine an extensive set of postulated predictor variables in a large, UK-based sample stratified by severity of persecutory beliefs. In random forests, several decision trees are created, and predictions (e.g., levels of persecutory beliefs) are calculated by averaging the individual predictions of all decision trees[13]. ML algorithms, such as random forests, offer several advantages over conventional statistical analyses. Specifically, they (1) are capable of handling datasets with a large number of predictors[13], (2) do not require assumptions regarding linearity, normality, and homoscedasticity, which are typically required in conventional regression analyses[14], and can (3) identify complex and non-linear relationships between predictor variables and outcomes without requiring pre-specifications[13,15,16]. These advantages make random forests a valuable tool when dealing with a large number of predictors.

## Methods

A systematic literature search identified existing aetiological models of delusions in general and/or persecutory delusions specifically. From these, predictors were extracted and assessed in a cross-sectional observational online study.

### Literature search and factor extraction

This study follows up on a theoretical review and synthesis of existing models of delusion formation[10]. As part of this review, a systematic search of the peer-reviewed literature published after 1990 in PubMed, ScienceDirect, and Web of Science was conducted by S.D. This search resulted in 51 publications that fit the inclusion criteria (see Fig. 1A)[17–67]. Two additional models were published after the data collection for the present study (i.e., refs. 68,69). From the 51 models, S.D. extracted predictors explicitly proposed to contribute to the formation of delusions in general and/or persecutory delusions specifically. As depicted in Fig. 1B, a total of 357 proposed

predictors were identified. By omitting repetitions, predictors were compiled into 66 unique predictors. Although the general principle was to include all predictors outlined in the models, a few exclusion criteria were set. Predictors were excluded for the following reasons: a) practical constraints, such as the length of the task or questionnaire, b) the absence of a reliable and validated method for assessing the variable, c) primary involvement in the maintenance of delusional beliefs rather than the formation, and d) substantial conceptual overlap with persecutory beliefs (i.e., in cases where the target variable was either identical to or a core component of the predictor, rendering the prediction tautological). A list of excluded predictors is provided in Supplementary Table S1. In total, 55 predictors that included social, cognitive, Bayesian, associative learning, and neurobiological aspects were included and operationalised using brief questionnaires or behavioural paradigms.

### Participants and procedure

We preregistered the study design and analysis plan with the OSF (https://osf.io/tuayf, 19th December 2022). Participants were recruited via the online recruitment platform Prolific (www.prolific.com). A quota sampling strategy was used to attain sufficient variability in the primary dependent variable (i.e., persecutory beliefs). The participant flow is displayed in Fig. 2. A gender-balanced pool of participants who had provided informed consent, were at least 18 years old, fluent in English, and residing in the UK were invited to participate in the online screening. Only participants with a history of providing good-quality responses (i.e., an acceptance rate of ≥95%) were invited to ensure high data quality. Eligible participants were able to access the screening through their Prolific dashboard. They were broadly informed about the study aims (i.e., "investigating associated factors of beliefs about the world and other people") and the procedure. In the screening, participants completed questions on their age, sex assigned at birth, self-identified gender, and persecutory beliefs (Revised Green Paranoid Thoughts Scale [R-GPTS] Persecution subscale)[70]. The median screening duration was 1 min ($M = 1.37$, $SD = 0.81$). Based on the screening, participants were assigned to one of four quotas according to their persecutory beliefs score (R-GPTS Persecution): average (≤5), elevated (6−10), moderate (11−17), and severe or very severe (≥18). A cut-off of 11 on the R-GPTS Persecution subscale has been proposed as the optimal cut-off for discriminating between patients with persecutory delusions and a non-clinical group[70]. In the screening sample, 23.4% ($n = 285$) of individuals met this cut-off and were assigned to the elevated (12.2%, $n = 149$) or the severe/very severe quota (11.2%, $n = 136$; see Fig. 1B).

If eligible, participants received invitations to the main survey within 24 h after they had participated in the screening. The median (Mdn) delay between starting the screening and the main survey was 9.29 h (median absolute deviation [MAD] = 11.71). The median delay was descriptively longer in the moderate quota (Mdn = 16.17, MAD = 15.16) than in the other three quotas (average: Mdn = 7.34, MAD = 8.62; elevated: Mdn = 9.39, MAD = 11.58; severe/very severe: Mdn = 8.51, MAD = 10.74). We obtained complete data from 336 participants in the main study (i.e., $n = 84$ per quota), in accordance with a power analysis for multiple regression with medium effect size ($f^2 = 0.15$) and high power ($\beta = 0.95$) using G*Power (version 3.1.9.7)[71]. The median duration of the main survey was 98 min ($M = 106.1$, $SD = 31.4$).

Both the screening and the main survey were implemented and presented using the web-based open-access platform PsyToolkit[72,73]. PsyToolkit is a viable method for running online questionnaires and response time paradigms with the same accuracy as laboratory-based assessments[74]. To account for order effects, the questionnaires, items within questionnaires, and paradigms were presented in a random order, except that all participants completed a demographic questionnaire as well as a cognitive test (i.e., the Cognitive Reflection Task[75,76]) at the end of the main survey. Since the Cognitive Reflection Task includes trick questions, this was necessary to avoid confusion or altered responses to the other questionnaires. Four attention checks were spread out randomly throughout the main survey to ensure attentive participation. Participants received

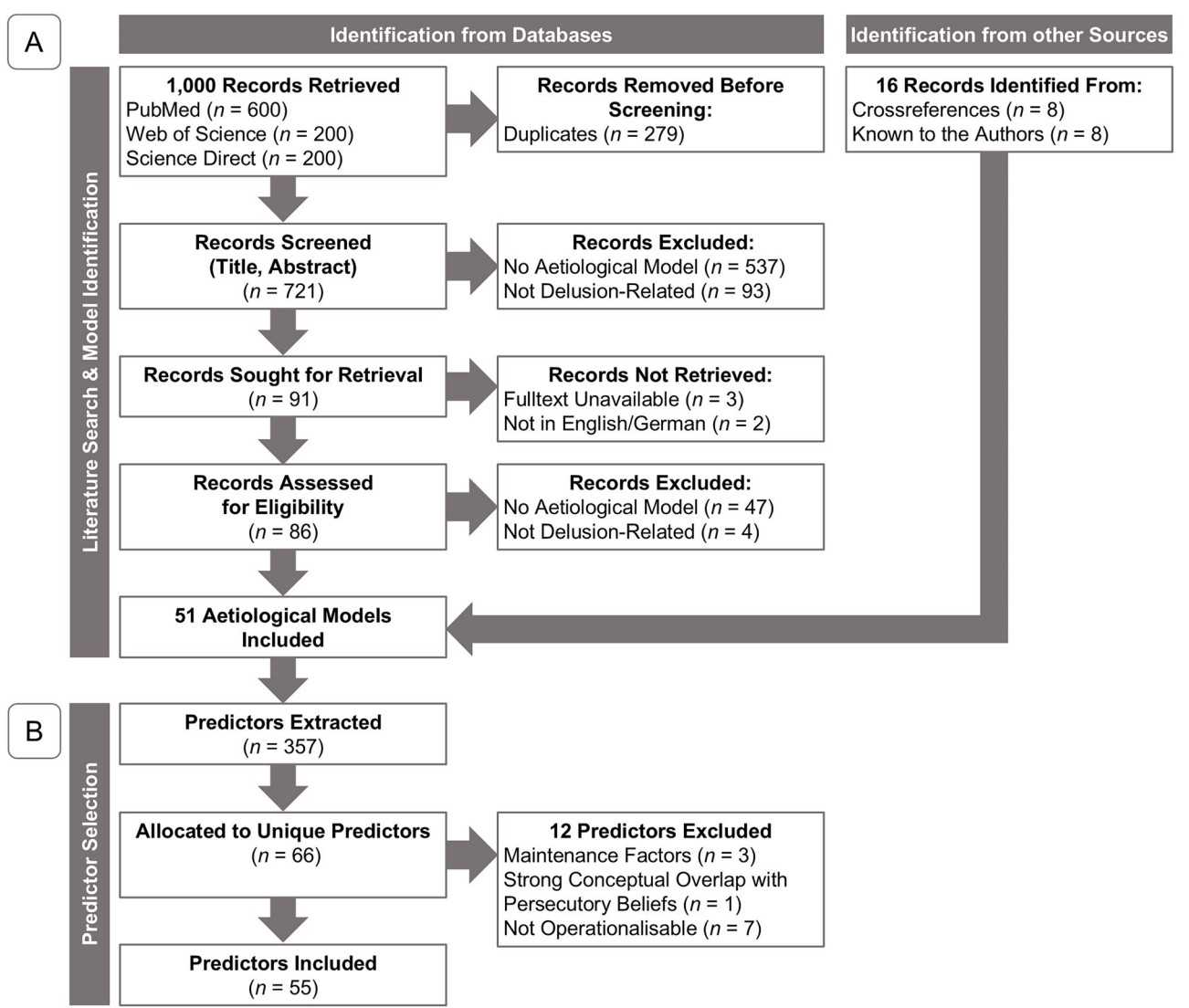

**Fig. 1 | Overview of the literature search and predictor selection. A** displays the literature search, and **B** provides an overview of the selection of the predictors. The literature search was conducted as part of a previous publication[10]. The first 600 records were retrieved from PubMed. These were supplemented with the first 200 records from Web of Science and the first 200 records from Science Direct. Eight models that did not appear in the literature search but were known to the authors and fit the inclusion criteria were included[17,32–34,45,59,61,63].

monetary compensation for their participation in the screening (£0.12) and main survey (£14) via their Prolific account. The procedure of this study was carried out following the Declaration of Helsinki (2008) and approved by the Local ethics committee of the University of Hamburg (2022_029).

**Measures**

**Predictor variables.** The self-report measures and behavioural paradigms used to assess the predictor variables are summarised in Table 1.

**Outcome variables.** As a measure of persecutory beliefs and ideas of reference, participants completed the R-GPTS[70]. The sum score of the Persecution subscale served as the primary target variable, namely persecutory beliefs. It consists of ten items assessing paranoid beliefs during the last week (e.g., "I was sure someone wanted to hurt me"), rated on a scale from 0 (not at all) to 4 (totally). Rated on the same scale, the Reference subscale consists of eight items (e.g., "People have been dropping hints for me"). The R-GPTS is frequently used in research on persecutory beliefs[77] due to its excellent psychometric properties. In our

sample, the Persecution and Reference subscales had an internal consistency of $\alpha = 0.90$ and $\alpha = 0.91$, respectively.

A broader range of delusional beliefs was assessed with the 21-item Peters Delusion Inventory (PDI)[78]. On dichotomous Yes/No items, participants indicated whether they experienced delusional beliefs (e.g., "Do your thoughts ever feel alien to you in some way?") and rated the associated distress on a scale from 0 (not at all distressing) to 4 (very distressing). In our sample, the PDI had an internal consistency of $\alpha = 0.88$.

We assessed general psychopathological symptoms utilising the Symptom Checklist K-9 (SCL-K-9)[79]. On nine items, it assesses the experience of diverse psychological symptoms (e.g., somatisation, anxiety, obsessiveness) during the past week on a scale from 0 (not at all) to 4 (extremely). In our sample, the SCL-K-9 demonstrated an internal consistency of $\alpha = 0.90$.

**Demographics.** We assessed participants' current or past diagnoses of mental disorders, and current psychiatric or psychological treatment (including pharmacotherapy, psychotherapy, and counselling). Prolific provided information on participants' self-identified ethnicity.

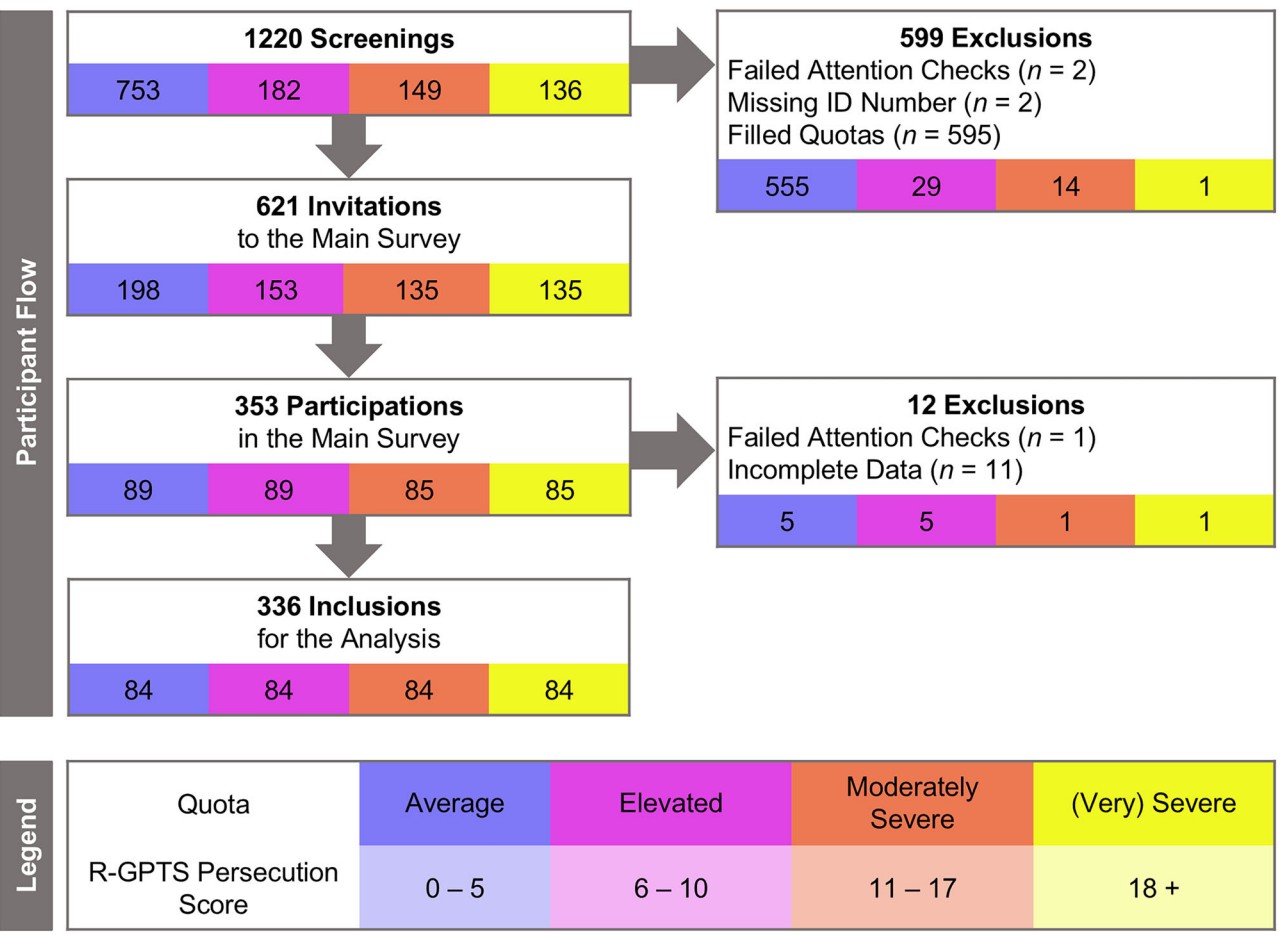

**Fig. 2 | Overview of the participant flow.** R-GPTS Revised Green Paranoid Thoughts Scale, ID identification. Quota thresholds are based on Freeman and colleagues[70].

## Statistical analyses

We utilised a machine learning (ML) approach with random forests to predict persecutory beliefs (R-GPTS persecution scores) using the predictor variables from the questionnaires and computer paradigms as independent variables. R version 4.4.3 was used for calculating sum scores and sample characteristics, and for preparing the prediction plots (Fig. 3). The *fmincon* function in MATLAB (version R2022b) was used for modelling the Bayesian inference variables (see Table 1, Beads estimation task). The random forests were built in Python 3.8.6 using the package *scikit-learn*[80]. Missing values on three variables (Likelihood weight: $n = 24$; Prior weight: $n = 24$; hinting task: $n = 1$) were imputed via k-nearest neighbour single-imputation using the non-weighted mean of the three nearest neighbours.

Random forests are an ensemble learning method that calculates a multitude of decision trees, each on a random subsample of a training dataset[13,14,81]. To generate predictions (e.g., levels of persecutory beliefs), individual decision tree outputs are averaged for metric outcomes. In the case of categorical outcomes, the random forest prediction constitutes the majority vote (e.g., high levels vs. low levels of persecutory beliefs) across all decision trees. We built our random forests in multiple steps. First, we used nested cross-validation with ten outer and ten inner folds (i.e., random splits of the data) to tune the hyperparameters (e.g., the number of trees in the forest) of the random forests and evaluate their performance. In each outer split, the data is first divided into an outer fold training and testing dataset. Then, the outer fold training dataset is partitioned further into ten inner folds, consisting of a training and validation dataset each. Various random forests with different hyperparameter configurations are trained on the inner fold training datasets and evaluated using the corresponding inner fold validation dataset. This approach enabled us to identify the best hyperparameter configuration that yielded the highest average performance

(i.e., with the highest explained variance [$R^2$]) across the inner folds. Once the optimal hyperparameter configuration is identified in the inner folds, a random forest is calculated on the entire outer fold training dataset using this optimal hyperparameter configuration. Subsequently, the trained random forest is applied to the outer fold test dataset to evaluate its performance. This procedure is repeated across all outer folds and is used to reduce the risk of overfitting.

We tested for all possible hyperparameter configurations during hyperparameter tuning (i.e., grid search) using the hyperparameter space (i.e., possible values of the hyperparameters) displayed in Supplementary Table S4. Subsequently, the best hyperparameter configuration (i.e., highest) was used to train the random forests on the respective outer fold training data. These random forests were then applied to the outer fold testing data to evaluate the explained variance of the prediction model (Q1). To test which variables best predict persecutory beliefs (i.e., R-GPTS Persecution; Q2), we calculated SHapley Additive exPlanations (SHAP)[82] during the nested cross-validation procedure. These SHAP values represent the individual contribution of each variable to the prediction (e.g., the contribution of one's anxiety score towards the random forest's prediction of the respective person having high levels of persecutory beliefs) and are calculated taking all other predictors and their presumably complex interactions into account. Thus, the ML models account for the overlapping predictive value between variables. We calculated the SHAP explainer function (i.e., *treeSHAP*) for each random forest prediction model in the outer fold training datasets and applied it to the respective outer fold testing dataset. This algorithm is optimised for tree ensemble methods and, therefore, provides accurate and fast estimations for random forests[83]. We

## Table 1 | Utilised Self-Report Questionnaires and Behavioural Paradigms

| Self-report Questionnaires | | | | | |
|---|---|---|---|---|---|
| **Predictor** | **Instrument** | **N° Items** | **Scale (Anchors)** | **Internal Consistency Cronbach's α** | **Example Item** |
| Aberrant Salience | Aberrant Salience Inventory (ASI)[110] | 10 | *Yes* (1)/ *No* (0) | 0.82 | "Do normally trivial observations sometimes take on an ominous significance?" |
| Anxiety | Depression Anxiety Stress Scales (DASS-21)[111] | 7 | 4-point (0-3) | 0.84 | "I felt scared without any good reason" |
| Cognitive Fusion | Cognitive Fusion Questionnaire-7 (CFQ-7)[112] | 7 | 7-point (1-7) | 0.96 | "I tend to get very entangled in my thoughts" |
| Depression | Depression Anxiety Stress Scales (DASS-21)[111] | 7 | 4-point (0-3) | 0.91 | "I felt that life was meaningless" |
| Dichotomous Thinking | Cognitive Biases in Psychosis Questionnaire (CBQp) - Dichotomous thinking subscale[113] | 5 | Three options (1–3) | 0.42 | "Imagine you are standing at a bus stop when the bus you have been waiting for drives past half-empty without stopping. I am most likely to think: A. People are always so nasty. B. People aren't very nice sometimes. C. The driver must be in a bad mood today" |
| Discrimination | Perceived Discrimination[114] | 7 | *Yes* (1)/ *No* (0) | 0.50 | "Have you ever been discriminated against due to your gender (e.g., being female)?" |
| Emotion Regulation | Difficulties in Emotion Regulation Scale (DERS-16)[115] | 16 | 5-point (1–5) | 0.95 | "When I am upset, I feel out of control" |
| Genetics | Screening based on the Family History Screen[116] | 2 | Yes (1)/ No (0) | - | "Did one of your family members ever get diagnosed with a mental illness (e.g., depression, anxiety disorder, schizophrenia spectrum disorder)?" |
| Guilt | Self-Report Instrument for the Assessment of Emotion-Specific Regulation Skills (ERSQ-ES)[117] | 1 | 5-point (0–4) | - | "During the past week, I felt guilty" |
| Hallucinations | Launay-Slade Hallucination Scale- Modified II (LSHS-II)[118] | 16 | 5-point (0–4) | 0.91 | "I have been troubled by hearing voices in my head" |
| Intolerance of Ambiguity | Intolerance of Ambiguity Scale[119] | 12 | 5-point (1–5) | 0.93 | "I can't stand being taken by surprise" |
| Life Purpose - Search | Meaning in Life Questionnaire (MLQ)[120] | 5 | 7-point (1-7) | 0.92 | "I am seeking a purpose or mission for my life" |
| Life Purpose - Presence | Meaning in Life Questionnaire (MLQ)[120] | 5 | 7-point (1-7) | 0.92 | "I understand my life's meaning" |
| Minority Group Status | Minority Status[114] | 5 | *Yes* (1)/ *No* (0) | 0.23 | "I belong to an ethnic minority group and/ or have a different skin colour than the majority of people living around me" |
| Negative Beliefs about Mistrust | Beliefs about Paranoia Scale (BaPS) - negative subscale[121] | 6 | 4-point (1–4) | 0.88 | "Thoughts about suspiciousness worry me" Note: The original items were rephrased from *paranoia* to *suspiciousness*, to include individuals with mild persecutory beliefs who might not identify their thoughts as paranoid. |
| Negative Other Schemas | Brief Core Schema Scales (BCSS)[122] | 6 | 5-point (0–4) | 0.93 | "Others are hostile" |
| Neurodevelopmental Hazards | Murray-Lewis Obstetric Complications Scale (MLOCS)[123] | 22 | *Yes* (1)/ *No* (0) | 0.49 | "Do you know of any of the following complications during your birthmother's pregnancy?" (e.g., "Inadequate foetal weight gain or lack of nutrients") |
| Ostracism | Ostracism Short Scale (OSS)[124] | 4 | 7-point (1-7) | 0.93 | "Others ignored me" |
| Overconfidence in Judgements | Beck Cognitive Insight Scale (BCIS) - self-certainty subscale[125] | 6 | 4-point (0-3) | 0.70 | "I can trust my own judgement at all times" |
| Perceived Control | Perceived Control Scale[126] | 5 | 7-point (1-7) | 0.90 | "I have little control over the things that happen to me" |
| Physical Health | Somatic Symptom Scale-8 (SSS-8)[127] | 8 | 5-point (0–4) | .81 | "Stomach or bowel problems" |
| Positive Beliefs about Mistrust | Beliefs about Paranoia Scale (BaPS) - survival subscale[121] | 6 | 4-point (1–4) | 0.89 | "It is important to be wary" Note: The original items were rephrased from *paranoid* to *wary*, to include |

## Table 1 (continued) | Utilised Self-Report Questionnaires and Behavioural Paradigms

**Self-report Questionnaires**

| Predictor | Instrument | N° Items | Scale (Anchors) | Internal Consistency Cronbach's α | Example Item |
|---|---|---|---|---|---|
| | | | | | individuals with mild persecutory beliefs who might not identify as being paranoid. |
| Self-awareness | Private Self-Consciousness Scale – Revised (P-SCS)[128] | 9 | 4-point (0-3) | 0.82 | "I think about myself a lot" |
| Self-beliefs - Positive | Brief Core Schema Scales (BCSS)[122] | 6 | 5-point (0–4) | 0.87 | "I am valuable" |
| Self-beliefs - Negative | Brief Core Schema Scales (BCSS)[122] | 6 | 5-point (0–4) | 0.85 | "I am unloved" |
| Sensory Deficit | Self-created items based on Capella-McDonnall[129] | 2 | 4-point (0-3) | 0.33 | "Do you have trouble seeing, even when wearing glasses or contact lenses?" |
| Sex | Assigned sex at birth | 1 | *Male* (0)/ *female* (1) | - | "Which sex was assigned to you at birth?" |
| Shame | Self-Report Instrument for the Assessment of Emotion-Specific Regulation Skills (ERSQ-ES)[117] | 1 | 5-point (0–4) | - | "During the past week, I felt ashamed" |
| Sleep Problems | Insomnia Severity Index[130] | 7 | 5-point (0–4) | 0.88 | "Within the past two weeks, I had difficulties falling asleep/staying asleep" |
| Social Anxiety | Brief Fear of Negative Evaluation (BFNE-II)[131] | 8 | 5-point (0–4) | 0.96 | "I am afraid that others will not approve of me" |
| Social Support | Interpersonal Support Evaluation List - short form (ISEL)[132] | 12 | 4-point (1–4) | 0.90 | "If I wanted to have lunch with someone, I could easily find someone to join me" |
| Socioeconomic Status | Income and education, as recommended by Lampert and Kroll[133] | 2 | 7-point (1-7) | 0.37 | "What is the average yearly income of your household?" |
| Stress | Depression Anxiety Stress Scales (DASS-21)[111] | 7 | 4-point (0-3) | 0.88 | "Over the past week, I found it hard to wind down" |
| Stress Reactivity | Arousal Predisposition Scale[134] | 12 | 5-point (1–5) | 0.87 | "I startle easily" |
| Substance Use | Substance Use Questionnaire (SUQ)[135] | 16 | 6-point (frequency: 1-6, quantity: 0-5) | 0.76 | "On average, how often did you consume alcohol in the past 3 months?" Note: Substance categories included alcohol, tobacco, cannabis, stimulants, opioids, hallucinogenic substances, misused medication, inhalants |
| Threat Anticipation | Negative Events Scale[136,137] | 10 | 7-point (1-7) | 0.92 | "How likely is it that the following things will happen to you in the future? Someone complains about your work" |
| Trauma | Childhood Abuse Questionnaire[138] | 4 | 6-point (0-5) | 0.81 | "Were you ever approached sexually against your will?" |
| Trust | Interpersonal Trust Short Scale (KUSIV3)[139] | 3 | 5-point (1–5) | 0.78 | "I am convinced that most people have good intentions" |
| Urbanicity during Childhood | City size – during childhood, based on Lincoln and colleagues[140] | 1 | 7-point (1-7) | - | "Until the age of 18, what best describes the area you lived in most of the time?" ("Less than 5.000 inhabitants" to "More than 10 million inhabitants") |
| Urbanicity at Present | City size – at present, based on Lincoln and colleagues[140] | 1 | 7-point (1-7) | - | "What best describes the area you currently live in?" ("Less than 5.000 inhabitants" to More than "10 million inhabitants") |
| Worry Thinking Style | 3-item Penn State Worry Questionnaire (PSWQ-3)[141] | 3 | 5-point (1–5) | 0.93 | "I worry all the time" |

**Behavioural Paradigms**

| Predictor | Paradigm | Trials | Description & Pre-processing |
|---|---|---|---|
| Attentional Bias Towards Threat | Emotional Stroop Task[142] | 48 | Participants are presented with 10 neutral (e.g., "chair") and 10 negative, threat-related words (e.g., "death") in four colours on the display. They are instructed to press one of four buttons on the keyboard corresponding to the word's colour as fast and accurately as possible. Participants completed 12 practice trials before the testing trials. Attentional bias towards threat was calculated by subtracting the mean reaction time (RT; in milliseconds) of neutral trials from the mean RT of threat trials for each participant. |
| Bias Against Disconfirmatory Evidence (BADE) | Bias Against Disconfirmatory Evidence (BADE) Task[143] | 16 | Participants are presented three consecutive sentences that describe a scenario. After each sentence they are requested to indicate which out of four interpretations of a scenario is most plausible on a scale from 0 (not |

**Table 1 (continued) | Utilised Self-Report Questionnaires and Behavioural Paradigms**

| Behavioural Paradigms | | | |
|---|---|---|---|
| **Predictor** | **Paradigm** | **Trials** | **Description & Pre-processing** |
| | | | plausible) to 10 (totally plausible): a correct interpretation (which appears unlikely based on the first sentence but gets increasingly more plausible based on the second and third describing sentence), two lure items (which appear plausible at first but then appear less plausible with more describing sentences), and an absurd item (which is unlikely at any point). The mean reduction of perceived plausibility of the lure items from statement 1 to statement 3 serves as index for BADE. |
| Externalising Bias<br><br>Personalising Bias | Internal, Personal and Situational Attributions Questionnaire (IPSAQ)[144] | 16 | Participants are presented with negative and positive scenarios (e.g., "A friend talked about you behind your back") and are asked to rate how much they attribute the event to a) themselves, b) another person, and c) the situation on scales from 0% to 100%. For each item and condition (i.e., a) internal, b) external, and c) situational), the rescaled percentage is calculated by dividing the percentage by the total given percentage for that item (i.e., condition percentage/ total percentage).<br>Following Mehl and colleagues[145], the externalising bias is calculated by subtracting the internal negative score from the internal positive score. The personalising bias is quantified by dividing the external negative score by the sum of the external negative and situational negative scores. |
| Intelligence | Hagen Matrices Test - 6 item version (HMT-S)[146] | 6 | Matrices with geometric shapes that follow logical rules are presented. Participants are asked to indicate which of six options follows logically from the given matrix. Participants complete two practice trials prior to the testing trials.<br>The number of correct answers serves as an indicator of fluid intelligence. |
| Jumping to Conclusions (JTC) Bias | Beads Task[147] | 3 | On three trials, participants are presented with a string of up to eight blue or red beads drawn from either one of two bowls. Bowl A contains 60% blue and 40% red beads, whereas bowl B contains 40% blue and 60% red beads. Participants are asked to indicate from which bowl the presented row of beads is most likely drawn. After each drawn bead, they can decide whether they are confident to make a decision and then indicate a bowl or whether they want further information (i.e., another bead).<br>The mean number of beads drawn during the three trials is taken as an index of jumping to conclusions. |
| Liberal Acceptance | BADE Task[143] | 16 | See *BADE* for a description of the task.<br>The plausibility rating of the absurd item at statement 1 was used as an index for Liberal Acceptance. |
| Likelihood Weight<br><br>Prior Weight | Beads Estimation Task[148] | 14 | We used an estimation version of the Beads Task[149] to estimate prior weight (i.e., how much weight is given to prior knowledge) and likelihood weight (i.e., how much weight is given to new information). This task differs from the original beads task (see JTC Bias) in that participants are asked to indicate the probability that the presented beads string was drawn from bowl A and bowl B on a scale from 0 (bowl A) to 100 (bowl B) after each drawn bead. Participants are then asked to make a final decision after the eighth bead.<br>To prevent early termination of the task and encourage participants to perform at their best, they were instructed that they would receive a bonus payment of 1€ based on the number of correct trials. After participation, the bonus payment was given to each person irrespective of their performance.<br>Using the *fmincon* function in MATLAB, we fitted a weighted Bayesian belief updating model with two free parameters (one prior weight $\omega 1$ and one likelihood weight $\omega 2$; previously reported by refs. 148,149) to the draw-by-draw probability estimates. Model fitting was performed for each participant to minimise the root mean squared error (RMSE) between the model-estimated probabilities and the participant's reported probability estimates. Participants' first estimate before the first bead draw was taken as the starting prior belief for a given trial. Data for sequences associated with an incorrect final decision were excluded from the analyses. Participants' scores with an error rate exceeding 32% (i.e., less than 10 correct trials) were excluded from the analyses ($n = 24$). For robustness, each participant's data was fit 100 times to the model, using random starting points between 0 and 20 for each free parameter (parameter bounds set to 0 and 20). The parameters associated with the iteration yielding the lowest RMSE served as the best-fitting parameters for the participant and were included as predictors in subsequent ML analyses. |
| Reasoning - Analytical<br><br>Reasoning - Intuitive | Cognitive Reflection Task (CRT)[75,76] | 8 | Participants were presented with eight short riddles (e.g., "A farmer had 15 sheep and all but 8 died. How many are left?") that are phrased to lure participants towards an intuitive but incorrect answer (i.e., $15 - 8 = 7$ are left) and a correct answer that requires some thought (i.e., eight are left). The number of correct responses serves as an index for analytical reasoning, whereas the number of intuitive answers indicates intuitive reasoning. Incorrect answers that are not intuitive are coded as 0. |
| Theory of Mind – Inferring intentions | Hinting Task[150] | 10 | Participants read social scenarios in which one character implies an intention without explicitly naming it (e.g., "George arrives in Angela's |

**Table 1 (continued) | Utilised Self-Report Questionnaires and Behavioural Paradigms**

| Behavioural Paradigms | | | |
|---|---|---|---|
| **Predictor** | **Paradigm** | **Trials** | **Description & Pre-processing** |
| | | | office after a long and hot journey down the highway. Angela immediately begins to talk about some business ideas. George interrupts Angela, saying: 'My, my! It was a long, hot journey down the highway!' What does George really mean when he says this?"). Participants are asked to describe what the character intends to say. Participants are then given a hint in the form of a second statement (e.g., "George goes on to say: 'I'm parched!'") and can change their answer or stay with the previously given answer. The Social Cognition Psychometric Evaluation (SCOPE) scoring criteria were used to quantify correct responses on the open-ended items[151]. The number of correct answers serves as an indicator of social cognition. |
| Theory of Mind – Inferring Mental States | Reading the Mind in the Eyes Test (short form)[152] | 10 | Ten black-and-white pictures of the eye regions of different individuals mimicking different emotions are presented. Participants are asked to indicate which emotion out of four options is displayed. The number of correct answers serves as an indicator of the ability to infer others' mental states. |
| Working Memory | Digit Span Backwards Task[153] | ≤16 | This task is based on the Wechsler Adult Intelligence Scale - Fourth Edition[153]. Per trial, numbers are presented in a given order. Participants are asked to repeat the presented numbers in reverse order. Trials start at two numbers and get longer after every second trial, with the longest trial involving eight numbers. The task is discontinued when two trials of the same length are answered incorrectly. The number of correct trials serves as an indicator of working memory function. |

*Note.* Internal consistencies are calculated as Cronbach's *a* based on the present sample.

further created a prediction plot, displaying the observed and predicted scores per person, to examine how well individual scores on persecutory beliefs were predicted by the model (Q3a).

In our preregistration, we had planned to conduct an iterative variable exclusion analysis to determine whether a more parsimonious model can explain persecutory beliefs similarly well compared to the complete model (Q3b). However, the full random forest model, including all 55 predictors, demonstrated relatively modest overall predictive power. In light of this result, we opted not to conduct the preregistered stepwise exclusion of predictors from least to most important, as we deemed the expected additional insights from such an analysis to be limited. While iterative exclusion can, in principle, improve model performance by reducing noise, we judged the analytic effort required for retraining and evaluating >50 ML models using nested cross-validation to be disproportionate to the likely gains in interpretability or predictive accuracy in this particular case. This deviation from the preregistered plan is transparently documented and acknowledged as a limitation.

Instead of the iterative approach, we conducted two exploratory analyses to judge the potential redundancy of additional predictors for predicting the continuous R-GPTS score. That is, we examined a) how much of the variance can be explained by only the ten variables with the highest SHAP values and b) how much variance is still explained without these ten variables when including the remaining 45 variables only. To further explore the predictive power of the models, we also explored the performance of c) a binary classification model predicting whether individuals meet the cut-off for clinically relevant persecutory beliefs (R-GPTS Persecution ≥11) and d) a multiclass classification model predicting the assigned four quotas (average, elevated, moderately severe, and severe or very severe).

Lastly, we evaluated the specificity of the model for persecutory beliefs (Q4). That is, we examined to what extent the model can similarly predict variance in persecutory beliefs and ideas of reference (R-GPTS Persecution and Reference), general delusional beliefs (Peters Delusion Inventory; PDI), and general psychopathological symptoms (Symptom Checklist K-9; SCL-K-9).

## Reporting summary

Further information on research design is available in the Nature Portfolio Reporting Summary linked to this article.

## Results

### Sample Characteristics

Demographic characteristics (gender, mean age, education level, ethnicity, marital status), diagnoses of psychotic or other mental disorders, and mean persecutory beliefs scores of the sample are displayed in Table 2. An overview of the reported mental health diagnoses per quota and means and standard deviations for all predictor variables are provided in Supplementary Tables S2 and S3, respectively. The absolute correlations between the predicted outcomes are shown in Supplementary Fig. S1.

### Q1: How much variance in persecutory beliefs is explained by the putative predictors?

The explained variance for the model predicting persecutory beliefs (i.e., R-GPTS Persecution) was 31%. The nested cross-validation results for each of the prediction models are depicted in Table 3. The exploratory binary classification model, predicting the cut-off for clinically relevant persecutory beliefs (R-GPTS Persecution ≥11)[70], resulted in an accuracy of 68% (see Supplementary Table S5). The exploratory multiclass classification model, predicting the four quotas (average, elevated, moderate, and severe or very severe), demonstrated an overall accuracy of 41%, with the highest recall (i.e., true positives) for the average quota (64%) and the lowest for the elevated quota (21%). The beeswarm plots displaying the ten variables with the highest SHAP values for these exploratory models can be found in the supplement (Figs. S6 and S7).

### Q2: Which predictors bear the highest relevance for explaining persecutory beliefs?

Figure 4A displays the feature importance (SHAP) analysis for the ML model predicting persecutory beliefs (i.e., R-GPTS Persecution). The following ten variables ranked highest for predicting persecutory beliefs: stress, emotion regulation difficulties, ostracism, threat anticipation, negative other beliefs, trust, aberrant salience, hallucinations, negative beliefs about mistrust, and cognitive fusion.

### Q3a: How accurately can individual scores of persecutory beliefs be predicted?

Figure 3 shows a scatterplot for the R-GPTS Persecution ML model that plots the predicted values against the observed values. Scatterplots for the

**Fig. 3 | Scatterplot and distribution of the observed and predicted R-GPTS Persecution scores.** $N$ = 336 participants; R-GPTS Persecution, Revised Green Paranoid Thoughts, Persecution subscale. The density plots represent the distribution of the observed (top) and predicted (right) R-GPTS Persecution scores. The predicted quota corresponds to the observed quota if the scores fall within the corresponding coloured areas.

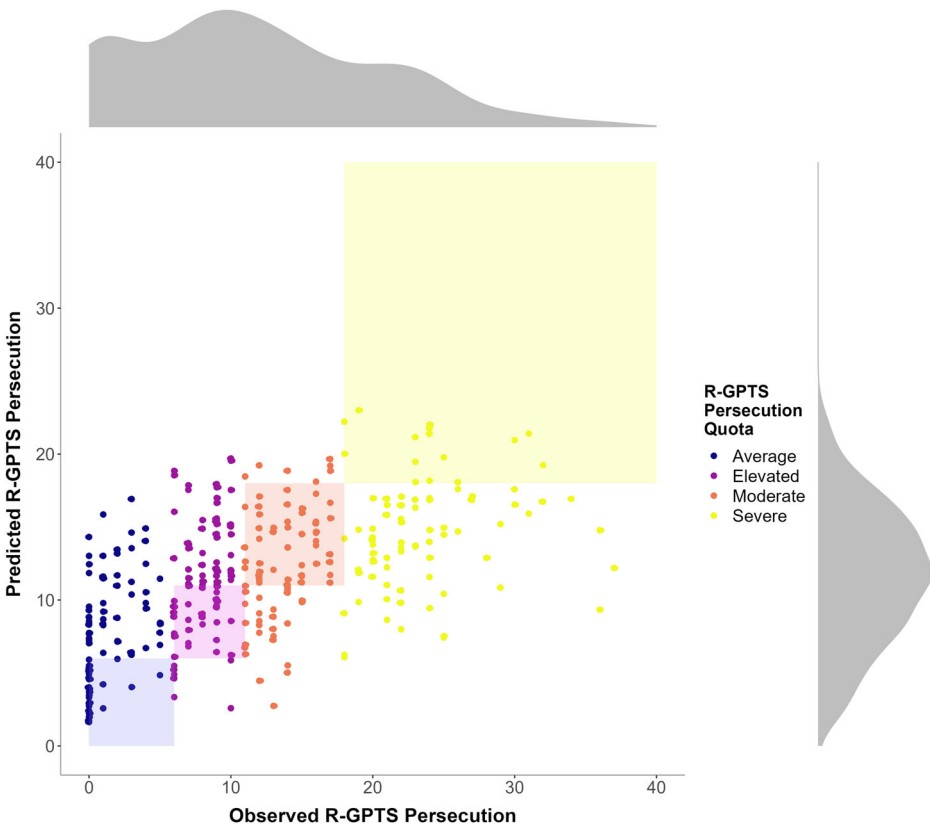

ML models predicting persecutory beliefs and ideas of reference, general delusions, and general psychopathology are depicted in Supplementary Fig. S2.

### Q3b: Can a reduced model predict variance in persecutory beliefs similar to the full model?

The model including only the ten highest-ranking variables explained 35% of the variance in persecutory beliefs. The reduced model, excluding these ten variables, yielded an explained variance of 22%. The scatterplots of the observed and predicted scores and the beeswarm plots displaying the ten variables with the highest SHAP values can be found in the supplement (Figs. S3 and S4).

### Q4: Are the putative predictors specific to persecutory beliefs rather than relevant to reference ideas, general delusions, or psychopathological symptoms?

As summarised in Table 3, the models predicted 53% of the variance in persecutory beliefs and ideas of reference (i.e., R-GPTS Persecution and Reference), 47% of the variance in general delusional ideas (i.e., PDI), and 77% of the variance in general psychopathological symptoms (i.e., SCL-K-9). Thus, the explained variance of the broader variables exceeded that of persecutory beliefs alone. Figure 4 displays the predictors with the respective highest relevance (i.e., SHAP value) for (A) persecutory beliefs, (B) persecutory beliefs and ideas of reference, (C) general delusions, and (D) psychopathological symptoms.

## Discussion

This study presents a systematic attempt to quantify the extent to which existing theoretical models of delusions explain variance in the continuum of persecutory beliefs. We assessed a broad range of proposed risk factors from >50 aetiological models. In a stratified sample, our ML model with nested cross-validation explained 31% of the variance in persecutory beliefs. In comparison, the ML models explained 53% of the variance in ideas of

reference and persecutory beliefs combined, 47% in delusions in general, and 77% in general psychopathology. Predictors with the largest influence on the persecutory beliefs model included metacognitive beliefs (negative beliefs about mistrust and cognitive fusion), social factors (ostracism, threat anticipation, generalised negative other beliefs, and trust), perceptual anomalies (aberrant salience and hallucinations), and negative affect (stress and emotion regulation difficulties). They thus stemmed primarily from models with a cognitive, social, and neurobiological perspective. A few predictors emerged as specifically influential for predicting delusional beliefs compared to predicting psychopathological symptoms in general. These distinctive predictors included ostracism, negative beliefs about others, threat anticipation, aberrant salience, and hallucinations.

The fact that nearly a third of the variance in persecutory beliefs is explained indicates that the factors postulated in aetiological models of delusions are indeed relevant contributors to predicting persecutory beliefs. It would be unreasonable to expect that we could fully account for the variance in highly complex psychological phenomena, such as persecutory beliefs, especially given the inherent limitations in measurement. At the same time, the relatively large proportion of unexplained variance suggests that the models are not capturing the complete picture. It seems that additional factors are necessary to account for the presence and severity of persecutory beliefs more fully. Alternatively, different compositions of predictors might be relevant to subgroups of individuals, leading to a reduced average explained variance. Prediction models based on multiple assessments over more extended periods of time would improve our understanding of both the directionality of associations and interactions between predictors and enable more personalised and improved predictions. Moreover, the limited explained variance, despite the large number of predictors, raises the question of whether some predictors have limited unique predictive value. To assess potential redundancy in our model, we estimated a reduced ML model that included only the ten highest-ranking variables. In this model, the ten predictors explained the same amount of variance as the complete set of predictors (i.e., 35%) - even slightly more -

**Table 2 | Socio-demographic Characteristics and Mean Persecutory Beliefs per Quota**

| Variable | Frequency (%) or Mean (SD) | | | | |
|---|---|---|---|---|---|
| | Persecutory Beliefs Quota (R-GPTS Persecution) | | | | |
| | Average[a] | Elevated[a] | Moderately Severe[a] | (Very) Severe[a] | Full Sample[b] |
| Gender (n, %) | | | | | |
| Male | 37 (44.0%) | 44 (52.4%) | 48 (57.1%) | 41 (48.8%) | 170 (50.6%) |
| Female | 47 (56.0%) | 36 (42.9%) | 35 (41.7%) | 43 (51.2%) | 161 (47.9%) |
| Nonbinary | 0 (0%) | 3 (3.6%) | 1 (1.2%) | 0 (0%) | 4 (1.2%) |
| None apply | 0 (0%) | 1 (1.2%) | 0 (0%) | 0 (0%) | 1 (0.3%) |
| Age (M, SD) | 44.2 (15.1) | 38.3 (12.3) | 38.7 (11.7) | 36.9 (11.3) | 39.5 (12.9) |
| Education (n, %) | | | | | |
| Primary School | 0 (0%) | 0 (0%) | 1 (1.2%) | 1 (1.2%) | 2 (0.6%) |
| GCSE or equivalent | 11 (13.1%) | 15 (17.9%) | 10 (11.9%) | 11 (13.1%) | 47 (14.0%) |
| A-level or equivalent | 15 (17.9%) | 12 (14.3%) | 15 (17.9%) | 19 (22.6%) | 61 (18.2%) |
| Higher National Certificate | 9 (10.7%) | 12 (14.3%) | 9 (10.7%) | 8 (9.5%) | 38 (11.3%) |
| Bachelor's Degree | 34 (40.5%) | 31 (36.9%) | 36 (42.9%) | 33 (39.3%) | 134 (39.9%) |
| Master's Degree | 13 (15.5%) | 12 (14.3%) | 11 (13.1%) | 9 (10.7%) | 45 (13.4%) |
| Doctorate or PhD | 2 (2.4%) | 2 (2.4%) | 2 (2.4%) | 3 (3.6%) | 9 (2.7%) |
| Diagnosis of a Psychotic Disorder (n, %) | 0 (0%) | 0 (0%) | 0 (0%) | 2 (2.4%) | 2 (0.6%) |
| Diagnosis of any other Mental Disorder (n, %) | 19 (22.6%) | 29 (34.5%) | 24 (28.6%) | 38 (45.2%) | 110 (32.7%) |
| Receiving Mental Health Treatment (n, %) | 12 (14.3%) | 12 (14.3%) | 18 (21.4%) | 24 (28.6%) | 66 (19.6%) |
| Ethnicity (n, %) | | | | | |
| Black | 1 (1.2%) | 1 (1.2%) | 4 (4.8%) | 0 (0%) | 6 (1.8%) |
| Asian | 6 (7.1%) | 1 (1.2%) | 5 (6.0%) | 6 (7.1%) | 18 (5.4%) |
| White | 75 (89.3%) | 79 (94.0%) | 68 (81.0%) | 71 (84.5%) | 293 (87.2%) |
| Mixed | 1 (1.2%) | 2 (2.4%) | 5 (6.0%) | 5 (6.0%) | 13 (3.9%) |
| Other | 1 (1.2%) | 0 (0%) | 2 (2.4%) | 0 (0%) | 3 (0.9%) |
| Unknown | 0 (0%) | 1 (1.2%) | 0 (0%) | 2 (2.4%) | 3 (0.9%) |
| Marital Status (n, %) | | | | | |
| Married | 25 (29.8%) | 27 (32.1%) | 21 (25.0%) | 27 (32.1%) | 100 (29.8%) |
| Divorced/ Separated | 9 (10.7%) | 3 (3.6%) | 8 (9.5%) | 4 (4.8%) | 24 (7.1%) |
| In a Relationship | 19 (22.6%) | 35 (41.7%) | 26 (31.0%) | 22 (26.2%) | 102 (30.4%) |
| Single | 29 (34.5%) | 19 (22.6%) | 29 (34.5%) | 30 (35.7%) | 107 (31.8%) |
| Unknown | 2 (2.4%) | 0 (0%) | 0 (0%) | 1 (1.2%) | 3 (0.9%) |
| Persecutory Beliefs (R-GPTS Persecution) (M, SD) | 1.3 (1.7) | 8.1 (1.4) | 13.7 (1.9) | 23.6 (4.4) | 11.7 (8.6) |

[a]$n$ = 84, [b]$N$ = 336.

**Table 3 | The Nested Cross-Validation Results of the Prediction Models**

| Model | $R^2$ | RMSE | MAE | Theoretical Scale Range |
|---|---|---|---|---|
| Persecutory Beliefs (R-GPTS Persecution) | | | | |
| Full Model (including 55 predictors) | .31 | 7.13 | 5.71 | 0 - 40 |
| Reduced Model (excluding the 10 most influential predictors) | .22 | 7.59 | 6.04 | 0 - 40 |
| Reduced Model (including only the top 10 most influential predictors) | .35 | 6.92 | 5.54 | 0 - 40 |
| Persecutory Beliefs & Ideas of Reference (R-GPTS) | .53 | 9.74 | 7.79 | 0 - 72 |
| General Delusions (PDI) | .47 | 10.86 | 8.14 | 0 - 84 |
| Psychopathology (SCL-K-9) | .77 | 3.68 | 2.94 | 0 - 36 |

*Note. R-GPTS* Revised Green Paranoid Thoughts Scale, *PDI* Peters Delusion Inventory, *SCL-K-9* Symptom Checklist K-9, *RMSE* root mean squared error, *MAE* mean absolute error.

likely due to increased statistical power. This finding questions the necessity of the additional 45 predictors, highlighting the particular relevance of the ten predictors in explaining persecutory beliefs. However, this does not necessarily mean that these specific ten predictors are uniquely crucial. Due to potential overlaps in predictive value, a different combination of predictors might yield similar results. To explore this, we ran a second reduced ML model, excluding the top ten predictors. The remaining 45 predictors still explained 22% of the variance, at least partially compensating for the predictive value of the top ten. These findings suggest that while the ten highest-ranking predictors were the most effective in explaining variance in the extent of persecutory beliefs in this sample, other predictors still contribute meaningfully and should not be disregarded entirely.

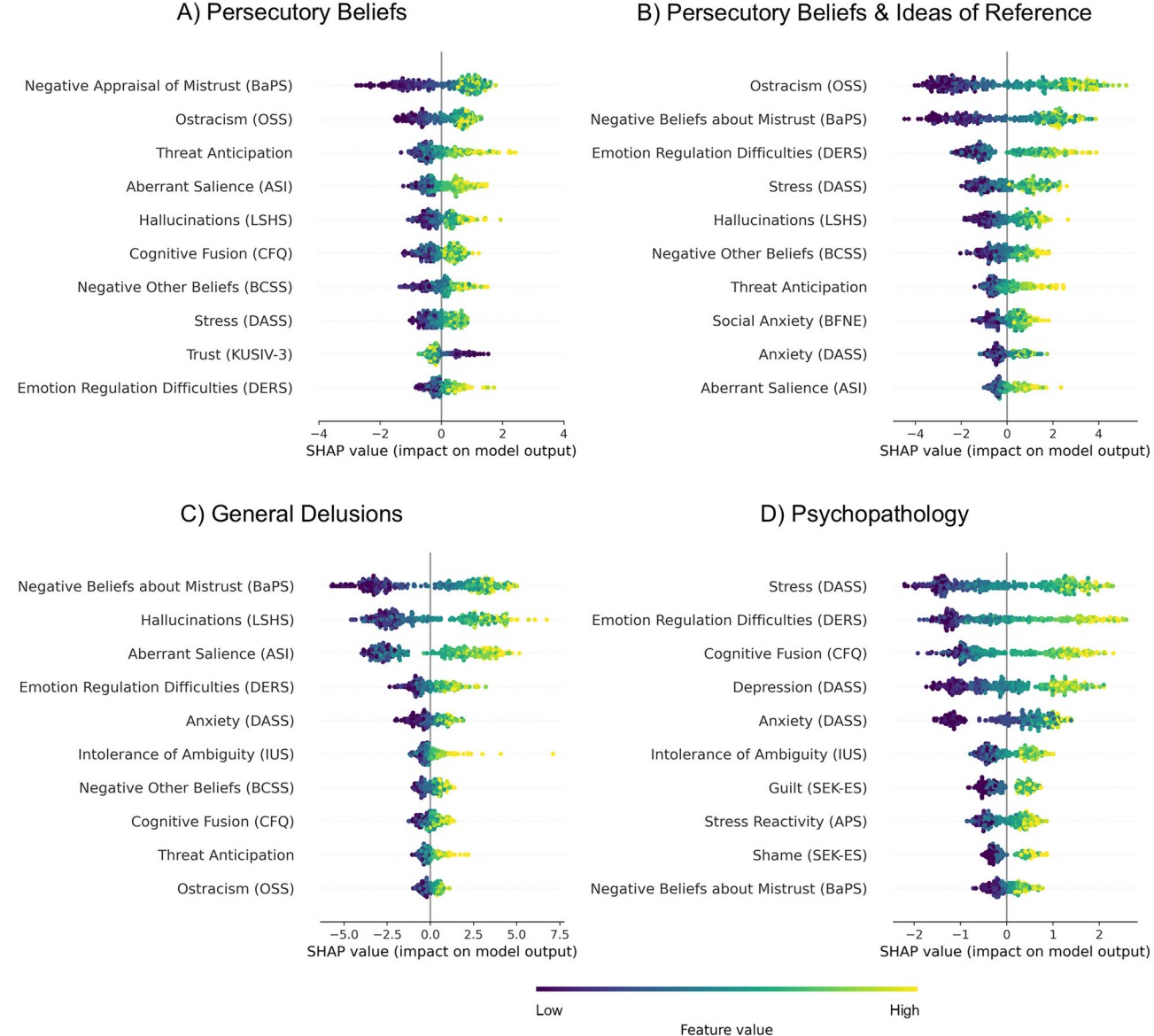

**Fig. 4 | Beeswarm plots of the ten highest-ranking predictors per machine learning model.** *N* = 336 participants. The plots display the ten predictors with the highest SHAP values for (**A**) the R-GPTS Persecution, (**B**) the R-GPTS Persecution and Reference, (**C**) the PDI, and (**D**) the SCL-K-9 in descending order. Each dot represents a participant, with its position on the *x*-axis indicating how much that variable influences the model's prediction of the individual score. When several dots overlap at the same position, they stack up to illustrate density. The colour of each point indicates the value of the predictor variable (i.e., feature value). Positive SHAP values indicate a change in the expected model prediction towards higher levels of the outcome variable; negative SHAP values indicate a change in the expected model prediction towards lower levels of the outcome variable.

To explore the robustness of our findings, we further built two exploratory models: a binary classification model predicting whether participants scored above the clinical cut-off on the R-GPTS Persecution, and a multiclass classification model predicting the assignment to one of the four quotas. The binary model demonstrated a relatively high accuracy (68%), suggesting that it was able to differentiate between individuals with clinically relevant persecutory beliefs and those without, with reasonable precision. Notably, the ten most influential predictors, as measured by SHAP values, closely matched those identified in the continuous model for persecutory beliefs, indicating consistency in the relevance of these predictors. The multiclass model predicting the four quotas performed less well, achieving an overall accuracy of only 41%. The model demonstrated the highest recall (i.e., true positives) for the average quota (64%) and the lowest for the elevated quota (21%). The overall reduced performance may be attributable to low statistical power due to the small sample sizes within each quota. Thus, these exploratory results should be interpreted with caution. The stronger performance of the binary model indicates that the predictors are more effective at distinguishing between individuals with and without clinically relevant persecutory experiences than at predicting the more granular levels of the extent of persecutory beliefs, such as intensity or frequency. This suggests that additional predictors (e.g., maintenance factors, external circumstances) are needed to predict the extent of paranoid symptomatology. Nonetheless, the solid performance of the binary model supports the usefulness of the proposed predictors in identifying individuals with elevated persecutory beliefs.

Compared to persecutory beliefs, the ML models explained more variance in the combined R-GPTS Persecution and Reference subscales and in the more broadly defined delusions in general. This could result from broader constructs being more easily explained compared to more narrowly defined, specific symptoms. Alternatively, it could be due to a better alignment of the combined R-GPTS scale to the target construct. Specifically, a recent study found the Persecution subscale to be more closely

related to clinician-rated functional deficits, whereas the Reference subscale was more closely related to clinician-rated persecutory ideas[84]. Accordingly, it has been recommended that both subscales should be used when examining the relationship between persecutory beliefs and their putative predictors or related factors[84]. Considering this, the relatively poorer performance of the model predicting persecutory beliefs alone may not reflect a failure of the proposed predictors to explain persecutory beliefs, but rather a better suitability of the combined scale to capture the construct more fully.

The model predicting general psychopathological symptoms performed even better than the delusion-focused models. The strong model performance as such is not entirely surprising, given that some of the psychopathology items overlap with included predictors (e.g., anxiety, depression). Moreover, many of the proposed risk factors, such as negative affect (i.e., anxiety, depression, stress) and emotion regulation difficulties, are known to be transdiagnostically relevant[85,86]. What was unexpected, however, is that the model prediction worked substantially better for general psychopathology than for persecutory beliefs, despite being theoretically tailored to the latter. This finding might support a hierarchical or dimensional model of psychopathology, wherein persecutory beliefs represent one specific expression of a higher-order general psychopathology factor. For instance, the Hierarchical Taxonomy of Psychopathology (HiTOP) proposes unusual beliefs, including persecutory beliefs, as one component of a thought disorder spectrum within a psychosis superspectrum[87]. In this framework, many of the predictors might be acting as indicators of a core liability, whereas persecutory beliefs only emerge when additional conditions are met, thus limiting their predictability compared to the broader construct.

Interestingly, individual scores within the severe end of the continuum of persecutory beliefs appeared to be less well-predicted by the ML model than the scores within the middle or lower end of the continuum (see Fig. 3). In particular, judging from the prediction plot, the ML model seemed to underestimate the highest scores in persecutory beliefs. One potential explanation is that since participants were informed that the study was about "factors associated with beliefs about the world and other people" before the screening, they may have intentionally scored higher on the R-GPTS Persecution subscale to be invited to the main survey. However, the predictions of persecutory beliefs with ideas of reference and of delusions in general appeared similarly poorer at higher severity levels (see Supplementary Fig. S2). Since both the R-GPTS Reference subscale and the PDI for general delusions were only assessed after the screening procedure, we argue that the poorer prediction of higher scores is unlikely due to a reporting bias. Taken together, the predictions seem to work best when the aim is to predict whether someone does or does not experience persecutory thoughts. In contrast, the predictions are unsatisfactory when it comes to predicting where exactly a person's score will fall, and this seems to be even more the case the higher we move up the continuum.

Two previous studies examined the explained variance in persecutory beliefs based on Freeman and colleagues' theoretical model of delusions[24]. Freeman, Dunn and colleagues[11] examined a subset of variables from their cognitive model in an undergraduate sample with limited variance in persecutory beliefs. Their regression model, including anomalous experiences, emotional processes (i.e., anxiety, depression, stress, self-focused attention, and social anxiety), and the need for closure, explained 44% of the variance in persecutory beliefs, ideas of reference, and depressive thoughts (i.e., Paranoia Scale[88]). When including ideas of reference, the variance explained by our ML model was comparably high (i.e., 53% explained variance) despite taking a more conservative statistical approach and using a stratified sample. A similar, more recent study set out to investigate the explained variance in persecutory beliefs (R-GPTS Persecution), using a set of predictors from the same cognitive model[24] (i.e., worrying, defence and avoidance behaviours, negative self-beliefs, aberrant salience, anxiety, discrimination, and social support) and additional predictors, including dissociation, analytical reasoning, and alcohol consumption[12]. Together, these predictors explained almost 67% of the variance in persecutory beliefs in a structural equation model. These results were not cross-validated or

corrected for multiple comparisons, which might have inflated the estimated explained variance. Further, the relatively larger variance in our sample, due to our stratified approach, likely contributed to the comparably lower explained variance in our ML model since the model was required to find patterns across the continuum of persecutory beliefs rather than in a more homogeneous low-severity group. Together with the applied cross-validation, our approach, therefore, provides a more robust estimate of the variance explained by aetiological models than previous studies.

Bearing in mind that the SHAP values take all other predictors as well as their interactions into account, several predictors emerged as particularly relevant for predicting persecutory beliefs and delusions. These included metacognitive beliefs (i.e., negative beliefs about mistrust and cognitive fusion), ostracism, hallucinations, and aberrant salience. The importance of perceptual anomalies such as aberrant salience and hallucinations in our ML model aligns with their relevance in the theoretical models of delusion aetiology. In a recent review, we found the formation of delusions as a means to explain an experienced anomaly or aberrant experience to be the shared key mechanism suggested by 53 existing aetiological models of delusions[10]. That is, models from cognitive, social, Bayesian, neurobiological, and associative learning perspectives all incorporated some form of aberrant perception in the aetiology of delusions. Recently, there has been a growing debate on whether aberrant perceptions, such as hallucinations, precede delusions or whether the mechanisms might be the other way around, with some studies supporting the latter[89,90]. Either way, aberrant salience and hallucinations had a high predictive value for persecutory beliefs and were specifically related to delusional beliefs rather than to transdiagnostic symptoms.

The high importance of metacognitive beliefs and ostracism for explaining both persecutory beliefs and ideas of reference stands in contrast to the weight they received in the aetiological models that were included (see Denecke and colleagues[10]). Only a few models emphasised the roles of metacognitive beliefs, such as negative beliefs about mistrust (e.g., perceiving suspiciousness as uncontrollable)[26,29,32] and cognitive fusion (e.g., the tendency to get very entangled with one's thoughts)[17,33] for delusion aetiology. These models suggest that individuals with higher cognitive fusion have trouble distancing themselves from their thoughts or considering that they can be erroneous, thereby promoting belief formation. They also assume that individuals high in the negative beliefs about mistrust will be more distressed by their thoughts, thereby promoting the recurrence of persecutory thoughts and beliefs[91]. Similar to metacognitive beliefs, social processes such as ostracism (i.e., feeling socially excluded) are rarely central to aetiological models of delusions. Some cognitive models[17,23–27,31,36,69] and a few social models[43,46,47] suggest that ostracism may contribute to persecutory beliefs. For instance, some social models interpret persecutory beliefs as an adaptive response to social threats such as discrimination or ostracism, emphasising the functional role of delusions rather than viewing them solely as deficits[47]. Since our data do not allow to infer directionality, we cannot entirely rule out the possibility that individuals with stronger persecutory beliefs scored higher on ostracism due to similarities in assessment items. However, experimental and experience-sampling studies support the idea that social exclusion can increase subsequent persecutory beliefs[92–94]. Given their relevance for predicting persecutory beliefs in our analyses, aetiological theories would benefit from more strongly recognising the importance of metacognitive beliefs such as negative beliefs about mistrust and cognitive fusion and social processes such as ostracism.

Interestingly, cognitive and reasoning biases did not have as much influence on our ML model as initially expected. Many aetiological models assume reasoning biases to play a central role in the development of delusions and persecutory beliefs (e.g.,[17,22,24,31]). Building on these models, treatment approaches focusing on reasoning, such as the metacognitive training for psychosis[95] have demonstrated small to medium effects on delusions by specifically targeting cognitive biases[96]. The relatively small contribution of reasoning biases to explaining persecutory beliefs in our study could be due to several factors. First, accurately measuring certain reasoning biases, such as belief updating, may require further validation and

standardisation for reliable assessment, as previous studies have varied widely in task and trial structure. Second, the contribution of reasoning biases might be underestimated due to shared method variance between predictors and outcomes measured via questionnaires. Third, online assessments of reasoning biases may be less reliable than those conducted in lab settings, potentially underestimating their contribution. However, previous online studies have found associations between persecutory beliefs and analytic reasoning and the BADE[97], as well as the JTC bias[98]. Our findings could thus suggest that reasoning biases, relative to other predictors, have a smaller explanatory value for the development of persecutory beliefs than initially assumed. In line with this interpretation, findings on the relevance of JTC for delusion development are mixed[99].

### Limitations
When interpreting the findings, several limitations should be kept in mind. First, our data is cross-sectional, thus not allowing for causal inferences. Further, predictors were assessed with some delay due to the screening procedure. The delay between screening and main survey was <10 h on average and was approximately equal across the quotas, with descriptively longer delays in the moderate quota. Such short delays are unlikely to have influenced the variance of persecutory beliefs and the resulting model predictions. However, as the mechanisms of delusion development unfold over time, the long-term predictive value of variables might be underestimated. Longitudinal studies assessing the broad range of predictors are required to investigate the extent and directionality of associations further. Second, online assessments and self-reports are susceptible to biases such as lack of attention, self-selection, unawareness, and recall bias, which might be especially relevant when assessing delusional beliefs. However, participants in online studies have been found to be at least as diligent, if not more so, as participants in the lab[100], with participants from Prolific providing the highest data quality compared to other online platforms[101]. Further, previous studies demonstrated that psychotic symptoms can be reliably assessed in carefully designed online studies[102], as well as by using patient self-reports[103]. Nonetheless, the individuals who participated in our study were literate, had access to the Internet, and were able to complete the relatively long survey, which may have been a boundary for individuals in acute episodes of psychosis. Moreover, persecutory beliefs are common in psychotic disorders, and one might have therefore expected a higher-than-average (0.30-0.66%)[104] prevalence of psychosis despite our sampling strategy not being designed to recruit individuals with a diagnosis specifically. However, only two participants (0.6%) reported a diagnosis of a psychosis-spectrum disorder. It is possible that such diagnoses were underreported due to stigma associated with schizophrenia-spectrum disorders[105,106] or that our sampling strategy was biased towards individuals without a diagnosis. Importantly, psychological distress was not generally low in our sample: participants in higher quotas of persecutory beliefs reported depression and anxiety disorder diagnoses as well as currently receiving mental health treatment more frequently than those in lower quotas. Nonetheless, whether the current findings replicate in samples with a higher prevalence of diagnoses of schizophrenia-spectrum disorders remains an open question that should be addressed in future studies. Third, a few predictors were not included in the study due to the lack of reliable and feasible assessments, our focus on the aetiology of delusions, and substantial conceptual overlap between predictors and persecutory beliefs (see Supplementary Table S1 for details). Although maintenance mechanisms were not included in this study, they may already play a role in the aetiology of persecutory beliefs by reinforcing and exacerbating threat beliefs once they occur. For instance, recent studies highlighted the role of safety behaviours in both the development and maintenance of persecutory beliefs[12]. Further, the decision of when a factor postulated in a model is conceptually too close to persecutory beliefs to be included remains somewhat subjective. We acknowledge that other criteria and boundaries for excluding predictors may also have been justifiable. However, even if the excluded variables had been included, they are unlikely to fully account for the remaining 69% of unexplained variance in persecutory beliefs. Fourth, despite our efforts, we cannot entirely rule out the possibility that some existing models of delusion formation that would have met our inclusion criteria were not identified in our literature search. Fifth, we deviated from our pre-registered analysis plan due to limited expected additional insight that would be gained from iteratively excluding predictors, given the already limited predictive power of the full model, which would be disproportionate to the high analytic effort of building over 50 ML models with nested cross-validation. Lastly, a higher shared methodological variance of questionnaires compared to computer paradigms might have contributed to differences in the relevance of predictors. Specifically, self-report measures may be more intercorrelated than correlated with computer paradigms due to the shared assessment method and its inherent biases. Since this is a common issue even in carefully designed studies using mixed assessment methods[107,108], whether and to what extent shared method variance influences the associations between predictors and delusions requires further examination. Including a behavioural measure of persecutory beliefs in addition to questionnaires may be beneficial to avoid this possible confound.

### Conclusions
Bearing these limitations in mind, our findings nevertheless indicate that a significant proportion of variance in persecutory beliefs across the continuum is not explained by the factors proposed to be relevant in contemporary aetiological models of delusions. This finding highlights the importance of examining, refining, and cross-validating theoretical models to reduce redundancy and improve our understanding of the aetiology of delusions. The relevance of metacognitive beliefs and ostracism for delusion development should be further investigated in longitudinal studies. Perceptual anomalies, such as aberrant salience and hallucinations, appear particularly important based on our findings as well as in theoretical accounts of delusion development. The explanatory value of potentially less predictive variables should be further examined and reevaluated. Discarding factors with little predictive value will help to develop more accurate and parsimonious theories of delusion aetiology. Moreover, future research could build on our results by examining how the predictors interact over time and influence delusional experiences in different individuals. Longitudinal and mechanistic studies, particularly those integrating experimental, neurocognitive, and experience-sampling methods, could clarify the causal pathways underlying delusions and refine aetiological models.

### Data availability
The de-identified and aggregated data for the reported analyses is available on the OSF[109] and is subject to a CC BY-NC-SA 4.0 licence.

### Code availability
The Python, Matlab, and R code replicating the current analyses are available on the OSF[109] and are subject to a CC BY-NC-SA 4.0 licence.

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

## Acknowledgements
We thank Brandon Ashinoff for his valuable suggestions on the Beads Estimation Task and for sharing the code for implementing the task and preprocessing the data. We further acknowledge financial support from the Open Access Publication Fund of the University of Hamburg.

## Author contributions
Conceptualisation: S.D., T.M.L.; Methodology: S.D.; Software: S.D., A.B.; Formal Analysis: F.S., S.D.; Data Curation: S.D., A.B.; Writing – Original Draft: S.D.; Writing – Review & Editing: S.D., T.M.L., A.B., F.S.; Visualisation: S.D., F.S.

## Funding

## Competing interests
The authors declare that there were no conflicts of interest.
