## [Transparent Peer Review file · Communications Psychology]

Using machine learning to predict persecutory beliefs based on aetiological models of delusions identified in a systematic literature search

Corresponding Author: Ms Saskia Denecke

Version 0:

Decision Letter:

Dear Ms Denecke,

Thank you for your patience during the peer-review process. Your manuscript titled "Using machine learning to predict persecutory beliefs based on aetiological models of delusions" has now been seen by 2 reviewers, and I include their comments at the end of this message. They find your work of interest but raised some important points. We are interested in the possibility of publishing your study in Communications Psychology, but would like to consider your responses to these concerns and assess a revised manuscript before we make a final decision on publication.

We therefore invite you to revise and resubmit your manuscript, along with a point-by-point response to the reviewers. Please highlight all changes in the manuscript text file.

Editorially, we consider it critical that all of the reviewers' methodological and conceptual concerns are comprehensively addressed. This includes the questions regarding the amount of predicted variance, concerns about the choice of prediction model selection and potential overlap between predictors and outcomes.

As you address the referee comments, please also ensure that your revisions comply with our preregistration policy - all previously preregistered analyses must be reported unless identified as unfeasible or fundamentally flawed (<https://www.nature.com/commspsychol/editorial-policies/preregistration-policy>). Finally, systematic literature analysis in the journal should follow PRISMA guidelines and include full PRISMA-compliant reporting and a PRISMA flowchart. Please see here: <https://www.prisma-statement.org/extensions> for appropriate PRISMA extensions.

I am attaching an Editorial Requests Table that details critical reporting requirements for the revised manuscript. Please attend to each item and ensure your manuscript is fully compliant. If your revised manuscript is not aligned with these requests on major issues, such as those concerning statistics, it may be returned to you for further revisions without re-review.

Please submit the following items:

- Revised manuscript
- Point-by-point response to the referees' comments
- Cover letter (as a separate document)
- <https://www.nature.com/documents/nr-reporting-summary.zip>>Nature Research Reporting Summary

- <https://www.nature.com/documents/nr-editorial-policy-checklist.pdf>>Editorial Policy Checklist
- Completed Editorial Request Table (attached).

via this link: Link Redacted .

Additional guidance is available in our style and formatting guide <https://www.nature.com/documents/commspsychol-style-formatting-guide-accept.pdf>>Communications Psychology formatting guide.

Best regards,

Erdem Pulcu

Dr Erdem Pulcu
Editorial board member
Communications Psychology
0000-0002-2170-0677

REVIEWER EXPERTISE:

Reviewer #1: schizophrenia, computational modelling
Reviewer #2: schizophrenia, computational modelling

REVIEWER REPORTS:

Reviewer #1 (Remarks to the Author):

I reviewed the manuscript "Using machine learning to predict persecutory beliefs based on aetiological models of delusions" submitted to Communications Psychology. In this manuscript, the authors had three aims: (1) estimate the accounted variance when predicting persecutory beliefs with different aetiological models, (2) identify the best predictors, and (3) test predicting specificity on persecutory beliefs against general delusions or general psychopathology. I believe they achieve the aims. Also, I first applaud the effort they did previously (2024 paper) to synthesize the aetiology in models of delusions. I see the current work as a follow up paper with an interesting ML approach to predict persecution given those variables. Although I have comments I hope can help improve the manuscript and its readability.

Main comment

Given the main analysis shows only 31% of explained variance I think the results could expand with alternative analysis. I would suggest testing alternative classification algorithms that could improve the 31% explained variance. If I understand correctly, you are predicting the continuous form of persecution, why not predicting categories (the 4 persecution levels; neural network), or even the paranoia clinical cut-off >11 (logistic regression). Perhaps with these alternative models you could gain more insights about how stable the 10 best predictors are and perhaps improve prediction. Currently the continuous persecution score is not well predicted (31%).

Secondary comments

Introduction

A clear introduction and straight to the point. A small comment. Given the predictors are based on the theoretical models you previously published, I believe it is worth to add a small paragraph in the introduction with a summary of your previous results (Denecke et al., 2024), so the readers know from where are you obtaining the predictors.

Methods: Literature Search and Factor Extraction

Make sure to clarify that the literature review was the aim of the previous Denecke et al. (2024) paper. Currently it feels that the literature review was specific for this paper.

Methods: Participants and Procedure

Link not working (<https://osf.io/tuayf>). The inclusion criteria to meet the severe quota is great, good balance in the groups, usually I find it difficult to recruit participants with high paranoia.

Measures

From the behavioural tasks you used, I only saw one task where a computational model was fit (Ashinoff, et al. JTC task). Maybe you can describe the reasons why no other computational models were used in other tasks. Also, several studies suggest that Reinforcement Learning (e.g., Probabilistic Reinforcement Learning task), Associative Learning (e.g., Kamin Blocking, or Latent Inhibition tasks), perceived animacy (e.g., Chasing Detection tasks) are closely related to paranoia, is there a reason why these behavioural paradigms were not used?

For curiosity, given that you have this in your database. If you only use questionnaires are these better than when you only use behavioural paradigms? There has been a recent debate open between questionnaires versus implicit measures (behavioural tasks).

Discussion

You mentioned "The latter is partly unsurprising given that some of the psychopathology items 290 overlap with included predictors (e.g., anxiety, depression)." Agree, is not surprising, but is it possible to conduct some sensitivity analysis? For example, did you explore predicting SCL-9 with predictors that remove the general psychopathology, like stress? Perhaps a further analysis to reduce the correlation between general psychopathology as predictor and general psychopathology as outcome.

Reviewer #2 (Remarks to the Author):

Hello – congratulations on an impressive piece of work. I recommend it for publication, with some minor revisions. Since my review is very long, for convenience I have extracted from it, and placed in a numbered list of 'action points', all the revisions/clarifications/comments I request of you. (In addition to these, within the review itself there are a couple of other 'suggestions' that you don't have to act on at all – one is a request for more detailed reporting of secondary analyses that would be nice but not necessary and may well be entirely unfeasible anyway; the other is merely a speculative comment that may be of interest and may be relevant to your overall conclusions, if you find it convincing, but which you might not want to pick up on and should equally feel free to just disregard. Since these aren't 'action points', you needn't feel obliged to respond to them and therefore they are not included in the numbered list below). The sub-section of the review within which the issue to which each action point below pertains is chiefly discussed, and/or from which the action point has been lifted directly as a quotation, is indicated for reference.

Action Points:

1) From 'General Summary and Evaluation' section: "The pre-registration has been followed faithfully, save for one planned analysis which the authors deemed not meaningful to proceed with in light of what they consider a lower-than-expected degree of predicted variance to have been achieved by all included predictors in the random forest analysis. On that point, it is not clear what the exact percentage of variance predicted was 'expected' a priori, nor why a failure to meet that threshold of predictive success (whatever it might be) would render the plan to perform iterative exclusion of predictor variables an uninformative analysis not worth proceeding with. Indeed, my understanding is that iteratively excluding uninformative predictors can frequently improve performance of random forest predictions - and in any case the results of such an analysis would have been useful for improving the interpretability of the central findings". ACTION: The precise technical reasons for which it was decided not to proceed with this planned analysis should be stated and explained, or (alternatively) the analysis should be conducted and reported as planned.

2) From 'Review of Scientific Aims and Methods' section: "It appears that for a cognitive task to qualify as a feasible instrument for inclusion in this study, it had to be quite short in terms of trial-number: I would have expected to see some reinforcement learning component in the study otherwise, and presumably the absence of any such task reflects the large number of trials that would be required to assay computational parameters meaningfully (in contrast to the more economic beads and JtC tasks, from which parameters can be estimated on the basis of a few trials corresponding to a short overall task duration)?" ACTION: Please comment on this briefly in response

3) 'Selection of Causal Models of Delusion Formation for Consideration' section: ACTION: "...please identify and justify the inclusion of these particular six models, and (if space allows) perhaps also comment on why the systematic review failed to identify some important areas of research (and on whether there are any other lacunae that are likely neglected in the final sample) or whether there were any other theories the authors noticed were likewise not picked up by the review procedure but which they ultimately deemed not as important to include as those six that they did decide it was essential to include, despite the review having failed to identify them."

4) 'Conceptual Overlap Between Predictors and Outcome' ACTION: this whole issue is by far the most serious concern I have about the paper, and I would ask that you engage with it not only in your response but also (since I think it likely other readers will be troubled by this too, if it is not addressed head-on) that you discuss it in the manuscript itself. This discussion should either respond to the concerns satisfactorily, or (equally acceptably and perhaps more feasibly) simply acknowledge

the issue as a limitation and consider what it might mean for the overall conclusions that can be drawn from the work (e.g. the greater significance, given their lack of conceptual overlap with the outcome and thus their relative disadvantage compared to most other top-performing predictors, of factors like 'aberrant salience' and 'hallucinations'). I feel strongly that this issue needs to be given discursive treatment in the text, but I do additionally suggest (within the relevant section of the review) a couple of ways in which more complete reporting of results could augment – though certainly I don't think it could replace – this verbal discussion. However, space and feasibility may not permit the fuller reporting of results I suggest, and in any case the issue needs to be tackled head-on in discussion, which need not refer to any results not already reported unless you feel it useful to do so. So long as you engage with the issue seriously in the text (either to recapitulate the points I have raised and acknowledge them – or else to introduce and somehow refute them and convincingly show why, after all, schizotypy alone ought to have been excluded) then I will be perfectly satisfied on this point.

5) From 'Conceptual Overlap Between Predictors and Outcome' section: ACTION: please describe the distribution (e.g. descriptive statistics and/or a figure illustrating this) of the temporal delay between screening and main experiment sessions, for each group. Comment on whether you think this temporal delay is likely to have been a concern or not

6) 'Improving the Introduction and Explanation of a Novel ML Method': ACTION: "At present, the cited references do not constitute anything approaching a sufficient reading list to supply the relevant background (they are neither numerous enough, introductory enough, or specifically germane enough to the point being made by the sentence within which they are cited) – many terms that could usefully be explicated in the main text (e.g. 'nested cross-validation', 'inner folds/outer folds') are left undefined, and the average reader will have to be motivated to independently undertake a great deal of investigative reading elsewhere, if they are to be in a position to infer the reasons for each of a number of very cursorily brushed over methodological choices in the details of this machine learning approach's implementation. Though I am satisfied that the methodological choices are suitable for the purposes to which they are intended (as a result of my having undertaken to do this additional research in the service of writing this review) there is a missed opportunity to make the paper's conclusions more intelligible, which could be rectified were its novel machine learning methodology more engagingly and transparently presented. I believe this would greatly improve the paper's impact (in every sense of that word). The novelty of the application of random forest machine learning prediction methods to delusion research is a strength of the work - but only to the extent that the majority of its probable readership is able to meaningfully engage with and appreciate the logic underlying it." I mention this general concern at several points throughout the review, and while it may sound like I am asking for a substantial rewrite, in fact I believe quite a little effort in the direction I have suggested would go a long way in making things easier and clearer for a reader new to these methods.

7) 'Screening Procedure and Interpretability of High Scores Absent Concomitant Diagnostic Prevalences': ACTION: "A couple of sentences further consideration of the issue would do wonders to reassure the reader about the relative insignificance of this potential issue or (equally acceptably) to highlight it as an actual, substantive limitation." Within the relevant section I suggest a couple of possibilities for the direction your response to this point might take, however of course you may take another tack entirely – all that matters is that you make some attempt to explain the anomaly in question and briefly explore its implications (or alternatively, that you demonstrate my concern to be entirely misguided,).

REVIEW:

General Summary and Evaluation:

This work is a novel and timely attempt to objectively compare a systematically curated set of the various causal factors that are implicated by current aetiological models of persecutory delusions (and that are, additionally, amenable to operationalization within an online population study) with respect to their importance as predictors of subjects' previously-obtained scores on a well-validated self-report measure of persecutory ideation (the R-GPTS questionnaire's persecutory subscale). In an initial online screening, members of the general population were administered this questionnaire and invitations to participate in the main online study were selectively issued in order to obtain a stratified sample that ensured equivalent representation of subjects within all levels of paranoia severity. This allowed random forest machine learning methods to be applied to the task of predicting paranoia severity on the basis of the included causal factors. The recruitment, sampling and testing methods are appropriate and their details have been judiciously decided with respect to the intrinsic trade-offs involved (trade-offs which the authors are cognizant of and which they thoughtfully grapple with in interpreting their findings). The discussion section is an excellent piece of scientific writing: its claims are clearly presented and cautiously evaluated, and the argumentation is convincing. However, I would like to see it expanded somewhat to address some concerns I have about the conceptual overlap that seems to exist between the target construct at issue and many of the included predictors thereof (particularly the most dominant predictors as per the random forest analysis's estimation of their importance). I also have some concerns about the interpretation of such high scores on R-GPTS measures of paranoid ideation as would typically be concomitant with clinical cut-offs for psychotic delusional illness, within an online quota sample in fact reporting very low prevalence of such disorders. Conceptual overlap of predictors and outcome, and the interpretability of scales intending to measure paranoia in this screened online sample, are issues that would benefit from some further consideration within the discussion section of the manuscript.

The random forest machine learning methodology employed to predict persecutory delusions and to estimate the importance of each of the included causal factors for generating these predictions, is generally well-suited for the intended purpose, and the particulars of its implementation seem well-chosen. Unfortunately, these analytic methods are not fully enough described and explained for this favourable conclusion to be apparent, without the reader first undertaking considerable additional independent reading to familiarize themselves with these approaches. Only by having done so, can readers who are not already familiar with random forest machine learning (such as myself, and presumably the majority of the paper's intended audience) understand what was done, and infer why it was done, in this study's analysis. This is a pity, as the novelty of the method's application (to a problem the authors rightly identify as hampering progress in the field of delusion research as a whole and which their methodology was deliberately chosen to address neutrally and comprehensively with respect to all

factions within that field) is a strength of the work. This strength ought to render the work more (not less) widely read and engaged with by delusion researchers – however, currently the conceptual basis of the analytic methods used are very underexplained, and the reach and readability of the work is likely to be reduced due to a missed opportunity to more didactically introduce the conceptual basis of the analytic methods (in recognition of its novelty and in anticipation of the epistemic needs of the large number of likely interested readers).

The pre-registration has been followed faithfully, save for one planned analysis which the authors deemed not meaningful to proceed with in light of what they consider a lower-than-expected degree of predicted variance to have been achieved by all included predictors in the random forest analysis. On that point, it is not clear what the exact percentage of variance predicted was 'expected' a priori, nor why a failure to meet that threshold of predictive success (whatever it might be) would render the plan to perform iterative exclusion of predictor variables an uninformative analysis not worth proceeding with. Indeed, my understanding is that iteratively excluding uninformative predictors can frequently improve performance of random forest predictions - and in any case the results of such an analysis would have been useful for improving the interpretability of the central findings. The precise technical reasons for which it was decided not to proceed with this planned analysis should be stated and explained, or (alternatively) the analysis should be conducted and reported as planned.

As a suggestion: if space permits and if the authors consider this feasible, it may be worth extending the supplementary material in order to report for all secondary and exploratory analyses (e.g. for the analysis omitting the first ten most important predictors) the same plots and level of detail as was afforded to the primary analysis, not just for completeness' sake but because these would be interesting in their own right (although again, I understand that it may be impracticable to implement this, so I suggest it merely as an option that for the authors to consider). On that note (and of some relevance to the issue of 'lower than expected' predictive success touched on above) in a later section of the review, I speculatively suggest (citing recent evidence) that, after all, the predictive success of the model – i.e. its predictive success evaluated with reference to the target construct of interest: paranoid delusions (as distinct from the instrument used to assay that construct) – may actually be less disappointing than the authors estimate it as having been. However, this idea is not in any sense proffered as something the authors need to include or engage with in their manuscript: it's not a suggested revision at all, merely a somewhat speculative pointer towards what could, if the authors judge that it would be beneficial to reframe things in this way, allow for a rather more optimistic evaluation of the degree of success with which they have managed to predict paranoia in the general population.

To summarise, I recommend this work for publication. At the same time, I suggest some minor revisions to the manuscript that I believe would improve its interpretability, and increase the reach and impact of the valuable work it reports. Aside from the general suggestion that the methodology be introduced and explained more fully, these suggestions mostly centre around the need to discuss certain outstanding conceptual issues (the most important of these being the 'conceptual overlap' issue detailed below, which I found particularly troubling), and in a few cases I request that the authors provide information about some specific detail if possible. My suggestion that the random forest methodology be introduced more fully may be quite accurately seen to merely reflect my own unfamiliarity with the approach (an approach which I should clarify that I have, as a prerequisite to reviewing this work, since familiarised myself with through careful background reading, and which I am as a result now confident I understand adequately – although not of course expertly). However, my position as a novice in this regard is hardly likely to be unique to (nor even atypical of) the readership of the paper, and indeed many psychological theorists of psychosis who do not regularly use computational methods of any sort in their own research may be even less well-positioned than myself to get by without further assistance on the part of the authors. I therefore hope that as a non-expert in machine learning, my advice (reflecting as it undoubtedly does my own ignorance) will be all the more useful for coming from the same epistemic position from which the paper will be approached by most of its readers. This is an excellent piece of work, and I enjoyed reading it and thinking about it. The minor revisions I suggest are intended to help the authors present it in a manner that does it justice, and that pre-emptively addresses what I envisage would be the most likely questions or concerns about it among a readership that I fully hope and expect will be a large and appreciative one.

Review of Scientific Aims and Methods:

The authors identify a problem for current research into the causes of paranoid delusion formation, which their study aims to address: despite the proliferation of aetiological models which continues apace, the field has failed to make definitive progress on explaining delusion formation anywhere concomitant with the calibre of the work being done in service of that agenda. That is, the question of what causes paranoid delusions – which everyone agrees has no unitary answer – ought to at least be answerable in terms of which of the many causal factors involved are major contributors (and which are relatively minor, or central only to certain paranoid presentations). No such ranking has been agreed upon because the development of causal models lacks constraint: there has been little in the way of systematic, impartial and scoping quantitative investigation into the relative predictive importance of the explanatory variables that are each central to one or another of the various models of paranoia's aetiology. As a result, the present Cambrian explosion of theoretical frameworks for understanding delusions formation lacks a concrete, externally-grounded basis for adjudicating between the vast array of aetiological models, in terms of which of these are the most likely to in fact explain a greater part of the overall variance in the observed phenomenon. This makes it difficult to prioritise the most promising avenues for future research, since to do so would require some viewpoint-agnostic, reliable metric for estimating the explanatory promise afforded by any given line of enquiry relative to its alternatives. The authors rightly note that there is not much direct competition between current theories of delusion formation – since each posits the central importance of some aetiological factor(s) that could - and (in large part at least) very likely do – causally explain some portion of the total variance, in who does and does not develop delusional beliefs, at a population level. The causes are complex and multifaceted and interactive, so adherents to one model have little independent reason to redirect their attention to another, absent some objective and universally-acceptable metric for gauging which specific factors appear to be particularly important in explaining variance in delusion formation within the population of interest.

The authors' aim in this paper is to provide such an impartial analysis: quantitatively estimating the relative importance of

causal factors implicated by current theories, with a view to guiding future research towards the most promising avenues of enquiry. This aim is a laudable one, and the attempt to offer a relatively 'neutral' metric for evaluating the merits of alternative theories using first a systematic review of the literature (to select candidate causes) and then machine learning (to rank their predictive importance) is well conceived. The population for which the authors thus attempt to predict variance paranoia severity is the general population, while the causal models included are garnered in the main from theories of how clinical paranoid delusions form (e.g. schizophrenia). Thus, the authors tacitly rely on a continuum approach to conceptualizing psychotic experience, in order to draw inferences from (A) "the relative importance of these factors for predicting paranoia severity in a non-clinical population", about (B) "the relative promise of the causal theories that invoke those factors to explain clinical paranoia". The contingent of clinically-oriented researchers who reject the notion of a meaningful continuum between the processes underlying non-clinical persecutory ideation and those underlying clinical delusions proper (or who insist only upon such a categorical distinction only when it comes to delusions in some particular subset of patients e.g. delusions in schizophreniform illnesses) will understandably find this work less than relevant to understanding anything about a clinical phenomenon that they deny exists on a continuum with the kind of persecutory ideation measurable by validated questionnaires designed for general population use. That said, the causes of persecutory ideation in the general population are undoubtedly worthy of investigation in its own right - even if they are categorically different in kind from all (or some) clinical forms of delusion. (Indeed, understanding the causes of delusions in different populations and examining the degree of overlap or divergence between them is information that would very relevantly contribute to resolving debates around continuum vs categorical approaches). Moreover, the majority of researchers do accept the utility of a continuum framework for thinking about delusional beliefs: for readers not opposed to the continuum approach in principle, the application of it here will be judged an acceptable one, as the study avoids some common pitfalls associated with the continuum approach by taking a number of careful methodological steps (e.g. attention checks peppered throughout the main experiment, and the choice of a questionnaire instrument which has been well validated with respect to clinical cut-offs in previous patient studies as a primary outcome measure).

At all stages (from selecting a set of candidate causal models, to determining which ones to include, to evaluating their predictive importance quantitatively) there is an admirable (and generally successful) effort to maintain objectivity (as befits the study's overall purpose and the problem it sets out to tackle) evident in their choice of methods:

- (1) an independent literature review of current theories of delusion formation, supplemented with six unidentified additional models which the authors were personally aware of and that they judged ought to be included, despite the literature review having failed to identify them using systematic methods
- (2) a selection process by which this pool of models was winnowed down to only those for which the construct invoked as aetiologically important could be measured in an online study by means of a well-validated instrument: very often a questionnaire, though not uncommonly a cognitive task - and occasionally a computational model of cognitive task performance. It appears that for a cognitive task to qualify as a feasible instrument for inclusion in this study, it had to be quite short in terms of trial-number: I would have expected to see some reinforcement learning component in the study otherwise, and presumably the absence of any such task reflects the large number of trials that would be required to assay computational parameters meaningfully (in contrast to the more economic beads and JtC tasks, from which parameters can be estimated on the basis of a few trials corresponding to a short overall task duration)? In one instance only (which seems not particularly uniquely deserving of its singular fate) a model was excluded because it was deemed to conceptually overlap to a significant degree with the target construct i.e. delusional thoughts, especially paranoid ones.
- (3) a random-forest machine learning analysis to determine how important were each of these measures, in terms of contributing usefully to predicting previously-collected paranoia severity scores, within a paranoid-thoughts-stratified general population sample (selected by quota for comparable representation within each of four paranoia severity brackets, from a broader participant pool screened on the R-GPTS paranoia subscale questionnaire for paranoia severity). Predictors were combined with one another stochastically, across a large number of individual decision tree algorithms (each with access to a different with-replacement subset and tasked with predicting R-GPTS paranoia (and, in subsequent analyses, R-GPTS paranoia + reference ideas; delusional ideation tout court as measured by the PDI-21; and finally general psychopathological symptomatology)).

Selection of Causal Models of Delusion Formation for Consideration:

The authors' aim to provide a relatively value-independent, neutral assay of current causal models of paranoia is a laudable one. To curate the set of candidate causal models that may be considered for inclusion, they surveyed the field as a whole by conducting a systematic literature review of current theories of delusion formation. This set of candidates was then supplemented with six unidentified additional models which the authors were personally aware of and which they believed (I have no doubt rightly) ought to have been included on the basis of their merits and importance, despite the literature review having somehow failed to identify them within the purview of its own systematic methods and pre-defined scope.

I have minor concerns about the 'value-laden' decision to include an additional six causal models that didn't crop up in the systematic review but which the authors knew ought to be included (I myself could have thought of other worthy additions - though I refrain from listing them here since the point is emphatically not that 'twelve rather than six' ought to have been appended to the systematically curated pool of candidates derived from the review!). Although I trust that the lucky six were genuinely (even if not uniquely) deserving of their inclusion post-hoc, I would be curious to know which six models these were - this information is currently not reported anywhere, and the manuscript should be revised to include it for transparency. However, no matter how strongly I would (once I knew which they were) tend to agree on the importance of the six models in question, that extra-procedural decision does rather open the authors up to challenge from any reader who finds a model to which they themselves are particularly attached to have been left out - "since the systematic review wasn't binding anyway, whither my pet model?". This somewhat undermines the commendable objectivity of the method of theory selection in the first place, which was to my mind one of the strengths of the research design - given that the study's intention is after all to provide some viewpoint-independent summary of the relative predictive importance of the various theories that comprise the current state of play within delusion research. I am tempted to start kvetching about the specific models I wish

the authors had included, but what I ultimately wish (and what I think would represent the only clean solution to the problem of candidate model selection) is rather that they had included nothing but what the systematic review identified, or alternatively (had they noticed that the methodology of the systematic review was inadequate to capturing the state of the field, given that some work they were aware ought to be included was not identified within it) that they had come up with an expanded set of search criteria to ensure a more thorough but still unprejudiced sampling of the field as a whole, and conducted a literature review in accordance with those updated criteria. This is a quibble for which the best correction, at this juncture, would be as follows: please identify and justify the inclusion of these particular six models, and (if space allows) perhaps also comment on why the systematic review failed to identify some important areas of research (and on whether there are any other lacunae that are likely neglected in the final sample) or whether there were any other theories the authors noticed were likewise not picked up by the review procedure but which they ultimately deemed not as important to include as those six that they did decide it was essential to include, despite the review having failed to identify them.

Conceptual Overlap Between Predictors and Outcome:

The exclusion of schizotypy, on the grounds that it shares too much conceptual overlap with the target outcome (i.e. paranoid thoughts), seems strange to me. While I am highly sympathetic to the logic itself, by that very logic it seems troubling that many of the questionnaires that were included – and, importantly, that ended up ranking high on overall predictive importance – seem to share at least as much conceptual overlap, if not more, with that target outcome than does schizotypy. In their very content, paranoid delusions exhibit a manifest and intrinsic correspondence with the tendency to endorse statements indicating that one is subject to “ostracism”, that one “anticipates threat”, that one distrusts others, and that one generally has negative beliefs about other persons. The commonplace (indeed, even typical) paranoid delusion that I am ostracised and threatened by a conspiracy of my fellows, merely because it is present, will by virtue of its very nature be reflected in my endorsements of such questionnaire items. It will therefore be strongly predicted by questionnaires that score such endorsements highly – but this does not at all imply that there is necessarily any aetiological type of relationship between (for example) the actual state of “ostracism” (as the questionnaire, and its interpretation in the paper, conceives of it) and my state of “paranoia” (which in fact is directly constituted by my falsely believing that I am ostracised). Nor is there any necessary aetiological relationship between “threat anticipation” and a prototypical presentation of paranoia which constitutively involves anticipated threat (and will therefore be associated with high scores on questionnaires measuring same). The fact that my delusional state happens to revolve around the notion that I am besieged by immanent threat (as paranoid delusions very generally do) does not necessarily result from any pre-existing tendency on my part towards elevated “threat anticipation” more generally – and in an experimental design that measures my paranoia temporally prior to the stage at which it then obtains my measures on all possible predictors of paranoia (it is not clear from the paper how much time before - though it would be good to know this!), then there is no reason to even suspect that my elevated threat anticipation temporally preceded, never mind caused, a paranoid delusion which essentially constitutes an elevated anticipation of threat, in and of itself. To make the point another way: people do not tend to get delusional beliefs that there is something wrong with their dopamine system, even though dopaminergic dysregulation happens to be a true cause of delusions in general; the infidelity of one’s romantic partner (while perhaps in some general sense an example of what could be termed a ‘stressor’ that marginally elevates one’s risk of psychopathology) is not in any meaningful sense a cause of delusions, but it is a common theme in their content and a predictive relationship would be entirely expected to exist between “severity of delusions (measured yesterday)” and “scores on questionnaires tapping into experiences of being cheated on and preoccupation with same (administered today)” as a result of that semantic overlap alone. By the same token, it so happens that paranoid delusions tend almost definitionally to have as their content concerns about threat anticipation, the conviction that one is ostracised, the experience of distrust in one’s fellows etc. But the conceptual overlap between the typical self-reported beliefs constitutive of the outcome variable (paranoia) and the putative causal factors thereof (pre-existing trait mistrust, social ostracism proper, etc - factors that, since maintenance factors are explicitly outside the scope of this study and insofar as possible excluded from it, presumably had to temporally predate the outcome’s emergence in order to count as candidate causes of that outcome) by itself entails that questionnaires probing such experiences administered in the main experiment will have some predictive relationship with paranoid delusions assessed during screening. This would hold true even were it (hypothetically) the case that there was zero causal relationship whatsoever between true ostracism/antecedent tendencies towards threat anticipation or mistrust/etc, and the formation of delusionally paranoid states. It is likely that if I surveyed people with a scale whose items asked about their experiences of victimisation by conspiratorial agents, there would be a predictive relationship between people’s scores and the severity of their paranoid delusional ideation – and that would be a genuine predictive relationship – however it would not constitute evidence of any aetiological relationship between “actual conspiratorial victimisation” (or for that matter, between “pre-existing trait levels of propensity towards belief that one is conspired against”) and “the delusional states which commonly have that narrative scenario as their content”.

Similarly, given what we know about the typical phenomenological form (as distinct from the content) of delusions, it is not surprising that items like ‘suspiciousness (i.e. the content of the belief itself) feels uncontrollable’ (from the scale measuring ‘negative beliefs about mistrust’), and items like ‘I tend to get very entangled in my own thoughts’ (from the scale measuring ‘cognitive fusion’), are important predictors of the presence of paranoid delusional states: these instruments pick up on experiences that are essential and intrinsic to delusional thoughts’ phenomenology.

It is noticeable that the causal models which crop up as the most important predictors are, in a high proportion of cases, those that are measured with an instrument which directly probes some or other example of the most core and typical aspects of the target outcome to be predicted (i.e. of paranoid delusions) – suggesting that the problem of conceptual overlap, on which grounds schizotypy was pre-emptively excluded from the model, may well be more widespread and pernicious than could be successfully overcome by excluding only schizotypy while leaving all these other, equally if not even more conceptually overlapping, instruments still in the set of included predictors. The relative importances of those

predictors remaining, when one omits the top ten most important predictors (of which ten only 'aberrant salience', 'hallucinations', 'stress' and 'emotional regulation' are not intrinsically constitutive of definitionally core and/or prototypical features proper to the target domain of persecutory delusions themselves) is not reported but may be of interest in this regard. Even more welcome an addition would be an analysis that excluded (as was done pre-emptively for schizotypy) all those factors that in a conceptual sense are inextricable from the phenomenology of the target domain itself. This is not to suggest that an important aetiological role for such conceptually-related predictors is unlikely to be genuine (indeed the causal models that prompted such scales' inclusion are rooted in empirical and theoretical grounds for believing that they are), but the deck is unfairly stacked towards overemphasizing the importance aetiologically of those predictors that so happen to be measured by instruments that load directly on the same target construct whose aetiology is at issue. A full ranking of the importances of all predictors included in each analysis reported would allow the reader more novel information upon which to cultivate their own judgment as to the extent to which (for example) the conceptual overlap of 'threat anticipation' with the typical content of persecutory delusions themselves is driving most, or only a little, of its importance as a predictor of that outcome – based on the overall dominance of those predictors that share such an advantage, relative to those that do not and whose aetiological significance would be accordingly better supported (though not deductively necessitated) by a relatively high ranking given their lower degree of conceptual relatedness with the outcome being predicted. In this regard, it is especially interesting that (as the domain-general neurocomputational theory of delusion formation motivating their inclusion as potentially important causal factors thereof would predict they ought to be) the only two predictors of persecutory delusional ideas that both: (a) share a high predictive importance in predicting delusions regardless of thematic domain (i.e. regardless of whether the content of those delusions is persecutory (narrowly defined), persecutory and referential, or broadly defined to encompass items relating to many more of the common delusional themes as in the PDI-21); and (b) at the same time are nevertheless not broader predictors of high importance to general psychopathology...are the predictors 'aberrant salience' and 'hallucinations'. The most influential theoretical framework implicating these experiences as a causally potent stimulus for inferential delusional formation (despite its 'explanationist' emphasis) would not posit such a clear division between 'aetiological' and 'maintenance' factors as is envisaged (for understandable pragmatic reasons) by the authors' approach, but the authors' conclusions that these factors especially stand out as potentially particularly important in aetiological terms (which are in their discussion acknowledged to not necessarily be as clearly distinct from 'maintenance' processes in reality as was by necessity assumed by methodological selection procedures) may well, on account of the foregoing considerations (and in light of them) be considered still yet more plausible and forceful.

Improving the Introduction and Explanation of a Novel ML Method:

As alluded to above, the methodology is likely to be novel to many, if not the vast majority, of researchers to whom this paper will be of direct interest. Though the authors can rely on computational methods being familiar to many (though probably not the majority) of these readers, the predictive 'black box' machine learning approach used here is of a quite different breed than the kinds of computational modelling approaches prevalent in research into the cognitive neuroscience of delusions. Nor will the universally high level of statistical literacy that should and can be assumed to characterise the audience for the paper at large render the methodology intelligible from the minimal descriptive treatment it is afforded here. This is a shame since an introduction to this novel methodology would likely be of great interest to many readers who will be encountering it here for the first time. Moreover, it is necessary to understand the methodology in much more detail than is provided or signposted by the paper, simply to be in a position to evaluate the logic of the paper's methods and the soundness of its conclusions. At present, the cited references do not constitute anything approaching a sufficient reading list to supply the relevant background (they are neither numerous enough, introductory enough, or specifically germane enough to the point being made by the sentence within which they are cited) – many terms that could usefully be explicated in the main text (e.g. 'nested cross-validation', 'inner folds/outer folds') are left undefined, and the average reader will have to be motivated to independently undertake a great deal of investigative reading elsewhere, if they are to be in a position to infer the reasons for each of a number of very cursorily brushed over methodological choices in the details of this machine learning approach's implementation. Though I am satisfied that the methodological choices are suitable for the purposes to which they are intended (as a result of my having undertaken to do this additional research in the service of writing this review) there is a missed opportunity to make the paper's conclusions more intelligible, which could be rectified were its novel machine learning methodology more engagingly and transparently presented. I believe this would greatly improve the paper's impact (in every sense of that word). The novelty of the application of random forest machine learning prediction methods to delusion research is a strength of the work - but only to the extent that the majority of its probable readership is able to meaningfully engage with and appreciate the logic underlying it.

Screening Procedure and Interpretability of High Scores Absent Concomitant Diagnostic Prevalences:

The possibility of artificial inflation of persecutory belief questionnaire scores by subjects during the screening is thoughtfully considered, and this section of the discussion is convincing in its arguments. However, there is a striking disparity between the psychometric cut-off corresponding to the "likely" presence of clinical delusions of persecution (11) and the total absence of self-reported diagnosed psychotic illness within the 'moderately severe' group (scoring 11-18). Moreover, there were only two psychotic illness diagnoses among the equally large 'severe' group in the sample, - which is even more puzzling given that the lower threshold for inclusion in this latter group (a score of 18) has been previously shown to constitute a 'cut-off [that] is unlikely to identify incorrectly an individual as having a persecutory delusion when they do not' (Freeman D, Loe BS, Kingdon D, et al. The revised Green et al., Paranoid Thoughts Scale (R-GPTS): psychometric properties, severity ranges, and clinical cut-offs. *Psychological Medicine*. 2021;51(2):244-253. doi:10.1017/S0033291719003155).

This puzzling disparity between the prevalence of psychotic disorders that would be expected in such high scorers, and the

very low rates of same actually reported within these groups, warrants reflection. If the scale were functioning here (in the present study) in roughly the same way as it did in the studies responsible for validating it and establishing its clinical cut-off points (i.e. if the scale in the present context were working simply 'as it should') then that would suggest a great number of participants within the upper two quadrants of severity can, with a high degree of confidence, be identified as individuals who very likely are suffering from undiagnosed and untreated clinical psychosis. If this is deemed the correct interpretation - was there any provision made in the ethics for contacting these individuals and signposting them to appropriate clinical attention/care?

Alternatively - if the very low rates of diagnosed psychosis self-reported by subjects are deemed concomitant with the true (low) underlying prevalence of that disorder, within the subset of the sample in question – then what does this imply about the mismatch between how the R-GPTS scale operates, and what it measures, in the present study's context - as compared to the contexts in which it was psychometrically validated and upon which the interpretation of it is grounded? One possibility is that other diagnoses (not captured by 'psychotic illnesses' as the term is used here, but themselves likely to be associated with delusional symptoms of these high levels of severity) were much more prevalent in the present sample than was psychotic illness itself – if the disparity between the rates of psychosis typical of such elevated R-GPTS scores, and the very low rates of diagnosed psychotic illness actually self-reported in these groups here, can be attributed to a high prevalence of such clinical diagnoses (e.g. borderline personality disorder), then perhaps this would be worth mentioning in the manuscript, as it would make sense of the otherwise-puzzling mismatch. This mismatch, left unreckoned with and unexplained, presently casts a rather tall shadow of doubt over the outcome measure's validity: suggesting that inflation of self-reported persecutory ideation in the screening phase was likely a particularly pernicious problem with the sampling strategy after all. A couple of sentences further consideration of the issue would do wonders to reassure the reader about the relative insignificance of this potential issue or (equally acceptably) to highlight it as an actual, substantive limitation.

Comment: Failure to Predict Paranoia, or Failure to Measure it?:

The persecutory subscale of the R-GPTS alone may be less suitable, as an instrument for the measurement of specifically persecutory delusions, when used in isolation as compared to when used in combination with the 'ideas of reference' subscale of that same instrument - or even perhaps compared to just the 'ideas of reference' subscale alone, which (in clinically high-risk persons, at any rate) seems to more purely single out persecutory delusions per se than does the so-called 'persecutory' subscale (which by contrast seems to better capture general social functioning deficits) (Williams, Trevor F., et al. "The reliability and validity of the revised Green et al. paranoid thoughts scale in individuals at clinical high-risk for psychosis." *Acta Psychiatrica Scandinavica* 147.6 (2023): 623-633.).

In this regard, the better performance achieved by the random forest for predicting the combined score of these two R-GPTS subscales (compared to its performance for predicting scores on the persecutory subscale alone) may be interpreted as indicative of the closer relationship between the target construct for which all predictors were selected on the basis of their aetiological contribution to under current models (i.e. with the principal outcome of paranoid delusions specifically and a measurement instrument that was here used to assay another more general construct (i.e. the secondary outcome measure of 'paranoia plus reference'). This secondary outcome measure may, in actual fact, have more psychometric relevance to the primary target construct that the study first and foremost intended to predict (and, therefore, that it may have succeeded in doing so, more successfully than the authors perhaps realised).

To put this another way, it may be that the poorer prediction of "paranoia" (as operationalised by the persecutory subscale of the R-GPTS in this study) compared to the prediction of scores on another instrument, that was used as a secondary outcome measure here (i.e. the combined R-GPTS persecutory and reference subscales), reflects not poor prediction of paranoia by the causal factors theorized to cause it, but better measurement of paranoia by the instrument that was better predicted in the present study than was the instrument used to measure paranoia itself. If so, this secondary measure's better-than-anticipated prediction by the curated aetiological factors in question is suggestive not of those factors' failure to specifically capture aetiological processes important in the primary outcome construct of interest (paranoia), but rather of the better suitability of the combined instrument (compared to the persecutory subscale alone) for measuring paranoia. This seems to be in the authors' favour, and might be worth considering in their evaluation of their model's success – an evaluation which is admirably thoughtful and measured, but may be permissibly rendered a little bit more optimistic in light of what I suggest in the present paragraph, if the authors consider the above suggestions persuasive and if they have a mind to revise that section to include the cited reference that supports them.

** Visit Nature Research's author and referees' website at  for information about policies, services and author benefits**

Version 1:

Decision Letter:

Dear Ms Denecke,

Your manuscript titled "Using machine learning to predict persecutory beliefs based on aetiological models of delusions" has now been seen by our reviewers, whose comments appear below. In light of their advice I am delighted to say that we are happy, in principle, to publish a suitably revised version in Communications Psychology.

We therefore invite you to revise your paper one last time to address the remaining concerns of our reviewers and a list of editorial requests. At the same time we ask that you edit your manuscript to comply with our format requirements and to maximise the accessibility and therefore the impact of your work.

EDITORIAL REQUESTS:

SUBMISSION INFORMATION:

OPEN ACCESS:

*** DATA AVAILABILITY:**

Link Redacted

Best regards,

Troby Lui

Troby Lui, PhD
Associate Editor
Communications Psychology

Dr Erdem Pulcu
Editorial board member
Communications Psychology
0000-0002-2170-0677

REVIEWERS' COMMENTS:

Reviewer #1 (Remarks to the Author):

The authors responded to all my queries. I am happy with the new manuscript. No more comments from my end. I look forward to see the paper available online.

Reviewer #2 (Remarks to the Author):

Dear authors,

Thank you very much for your reply, and for taking the time and care to implement such thoroughly satisfying revisions (to the manuscript/supplementaries) and thoughtful explanatory replies (in your rebuttal letter) as to fully address each of my comments, concerns and suggestions. Thank you also for your kind words of appreciation and helpful signposting! Since you have so admirably addressed all the points raised, all that remains on my end is to congratulate you on the new and improved version of your excellent paper, and to recommend it unreservedly to the editors for publication in its current form. Reviewing your work has been a pleasure, and I am sure that your contribution will be widely read and enjoyed, its very innovative and impressive stuff indeed. Congratulations and best wishes.

Universität Hamburg
DER FORSCHUNG | DER LEHRE | DER BILDUNG

Saskia Denecke

University of Hamburg

Clinical Psychology and Psychotherapy

Von-Melle-Park 5, 20146 Hamburg

30.06.2025

Dear Dr Pulcu,

Dear Reviewers,

We sincerely appreciate your feedback on our manuscript. Your valuable comments and constructive suggestions have allowed us to enhance the description of the methods and refine the discussion of the machine learning models. We have thoroughly accommodated your recommendations in our revised version. Adjustments are highlighted in the manuscript. Please find a point-by-point response to your comments below.

Reviewer #1 (Remarks to the Author):

I reviewed the manuscript "Using machine learning to predict persecutory beliefs based on aetiological models of delusions" submitted to Communications Psychology. In this manuscript, the authors had three aims: (1) estimate the accounted variance when predicting persecutory beliefs with different aetiological models, (2) identify the best predictors, and (3) test predicting specificity on persecutory beliefs against general delusions or general psychopathology. I believe they achieve the aims. Also, I first applaud the effort they did previously (2024 paper) to synthesize the aetiology in models of delusions. I see the current work as a follow up paper with an interesting ML approach to predict persecution given those variables. Although I have comments I hope can help improve the manuscript and its readability.

Response: Thank you for your kind feedback and for taking the time to review our manuscript.

Main comment

Given the main analysis shows only 31% of explained variance I think the results could expand with alternative analysis. I would suggest testing alternative classification algorithms that could improve the 31% explained variance. If I understand correctly, you are predicting the continuous form of persecution, why not predicting categories (the 4 persecution levels; neural network), or even the paranoia clinical cut-off >11 (logistic regression). Perhaps with these alternative models you could gain more insights about how stable the 10 best predictors are and perhaps improve prediction. Currently the continuous persecution score is not well predicted (31%).

Response: We agree that the 31% explained variance leaves one wondering whether alternative analyses would confirm this finding. We have further justified the use of the random forests ML approach and explained its advantages. While adhering to this preregistered approach, we have implemented your suggestion to test the prediction of the four persecution

levels and the clinical cut-off of the R-GPTS to assess the stability of our results. We briefly discuss their results and implications. The prediction of the clinical cut-off indeed worked better compared to the continuous approach, with an accuracy of 68%. The predictors with the highest influence on the model were almost identical to the predictors of the continuous model. However, the model predicting the four quotas did not perform as well, with an overall accuracy of 41%. This is likely due to the small sample size per quota and resulting lack of power. The results of this exploratory model should thus be interpreted with caution. A table summarising the models' performance and the corresponding beeswarm plots for these models can be found in the supplement (Table S5, Figures S6 and S7).

We added the following sections to the manuscript:

Methods: "To further examine the performance of the random forests, we conducted several exploratory analyses. First, we examined the relevance of the highest-ranking variables and the potential redundancy of additional predictors for predicting the continuous R-GPTS score. That is, we examined a) how much of the variance can be explained by only the ten variables with the highest SHAP-value and b) how much variance is still explained without these ten variables when including the remaining 45 variables only. Next, we examined the performance of c) a binary classification model predicting whether individuals meet the cut-off for clinically relevant persecutory beliefs (R-GPTS Persecution > 11) and d) a multiclass classification model predicting the assigned four quotas (average, elevated, moderate, severe or very severe)." (pp. 9-10)

Results: "The exploratory binary classification model predicting the cut-off for clinically relevant persecutory beliefs (R-GPTS Persecution \geq 11)¹⁹ resulted in an accuracy of 68% (see Supplementary Table S5). The exploratory multiclass classification model predicting the four quotas (average, elevated, moderate, and severe or very severe) demonstrated an overall accuracy of 41% with the highest recall (i.e., true positives) for the average quota (64%) and the lowest for the elevated quota (21%). The beeswarm plots displaying the ten variables with the highest SHAP values for these exploratory models can be found in Supplement 3 (Figures S6 and S7)." (p. 11)

Discussion: "To explore the robustness of our findings, we further built two exploratory models: a binary classification model predicting whether participants scored above the clinical cut-off on the R-GPTS Persecution, and a multiclass classification model predicting the assignment to one of the four quotas. The binary model demonstrated a relatively high accuracy (68%), suggesting that it was able to differentiate between individuals with clinically relevant persecutory beliefs and those without, with reasonable precision. Notably, the ten most influential predictors, as measured by SHAP values, closely matched those identified in the continuous model for persecutory beliefs, indicating consistency in the relevance of these predictors. The multiclass model predicting the four quotas performed less well, achieving an overall accuracy of only 41%. The model demonstrated the highest recall (i.e., true positives) for the average quota (64%) and the lowest for the elevated quota (21%). The overall reduced performance may be attributable to low statistical power due to the small sample sizes within each quota. Thus, these exploratory results should be interpreted with caution. The stronger performance of the binary model suggests that the predictors are more effective at distinguishing between individuals with and without paranoid experiences than at predicting the more granular levels of the extent of persecutory beliefs, such as intensity or frequency. This suggests that additional predictors (e.g. maintenance factors, external circumstances) are needed to predict the extent of paranoid symptomatology. Nonetheless,

the solid performance of the binary model supports the usefulness of the proposed predictors in identifying individuals with elevated persecutory beliefs.” (pp. 13-14)

“Taken together, the predictions seem to work best when the aim is to predict whether someone does or does not experience persecutory thoughts. In contrast, the predictions are unsatisfactory when it comes to predicting where exactly a person’s score will fall, and this seems to be even more the case, the higher we move up the continuum.” (p. 15)

Secondary comments

Introduction

A clear introduction and straight to the point. A small comment. Given the predictors are based on the theoretical models you previously published, I believe it is worth to add a small paragraph in the introduction with a summary of your previous results (Denecke et al., 2024), so the readers know from where are you obtaining the predictors.

Response: We adjusted the introduction accordingly by briefly discussing the findings of the previous paper and more clearly stating the relation between the two projects:

Introduction: “Numerous theoretical models aim to explain the aetiology of delusions in general and persecutory delusions specifically. As summarised in a recent review of 53 models of delusions¹⁰, these take on different perspectives and have emphasised different mechanisms while agreeing on a shared key mechanism. Specifically, the models propose that delusions are formed to explain an experienced anomalous event. Postulated predictors of delusions range from cognitive aspects (e.g., generalised beliefs about the self and other people and reasoning biases) over social factors (e.g., social exclusion and deficits in social cognition) and computational Bayesian inference variables (e.g., weighing of prior beliefs and new information) to neurobiological aberrations (e.g., neurotransmitter dysregulation) and factors related to associative learning (e.g., attentional biases and conditioning).” (p. 3)

“Therefore, to follow up on the theoretical summary in Denecke et al. (2024)¹⁰, this study aims to (Q1) provide an estimate of the variance [...]” (p. 4)

Methods: Literature Search and Factor Extraction

Make sure to clarify that the literature review was the aim of the previous Denecke et al. (2024) paper. Currently it feels that the literature review was specific for this paper.

Response: Thank you for bringing this to our attention. As suggested, we adjusted the “Literature Search and Factor Extraction” section to clarify the relation between the previous review and this project:

Methods: “This study follows up on a theoretical review and synthesis of existing models of delusion formation¹⁰. As part of this review, a systematic search of the peer-reviewed literature published after 1990 in PubMed, ScienceDirect, and Web of Science, was conducted by S.D. This search resulted in 51 included publications (see Supplement for details). Two additional models were published after the data collection for the present study (i.e., ^{17,18}).” (p. 5)

Methods: Participants and Procedure

Link not working (<https://osf.io/tuayf>). The inclusion criteria to meet the severe quota is great, good balance in the groups, usually I difficult to recruit participants with high paranoia.

Response: The OSF preregistration is now publicly accessible (<https://osf.io/tuayf>).

Measures

From the behavioural tasks you used, I only saw one task where a computational model was fit (Ashinoff, et al. JTC task). Maybe you can describe the reasons why no other computational models were used in other tasks. Also, several studies suggests that Reinforcement Learning (e.g., Probabilistic Reinforcement Learning task), Associative Learning (e.g., Kamin Blocking, or Latent Inhibition tasks), perceived animacy (e.g., Chasing Detection tasks) are closely related to paranoia, is there a reason why these behavioural paradigms were not used? For curiosity, given that you have this in your database. If you only use questionnaires are these better than when you only use behavioural paradigms? There has been a recent debate open between questionnaires versus implicit measures (behavioural tasks).

Response: Thank you for raising these questions. To assess the prediction by the theoretical models, we only included predictors that were explicitly named in the included 51 models. To our knowledge, perceived animacy was not part of these models. Regarding reinforcement learning mechanisms, only partial reinforcement was mentioned by some models but only as a maintenance - not a formation - mechanism. Since we focused on the formation of delusions, this predictor was not included. For associative learning processes, the models included conditional inattendance (e.g., a lack of blocking or latent inhibition) and correlation detection as predictors. Unfortunately, we did not come across paradigms that were both frequently used and sufficiently short to be implemented in the already extensive online survey. Thus, including these predictors was not feasible in this study. We acknowledge that this is a limitation:

Limitations: “Third, a few predictors were not included in the study since they lacked validated measures or could not be operationalised in an online study (see Supplementary Table S1 for details).” (p. 19)

Your second point is interesting. Since predictors are either measured by questionnaire or paradigm in our data, we cannot assess whether a better prediction is due to the alignment of the method of measurement or due to a higher predictive value of the predictor. To test this, we would need to have both a paradigm and questionnaire measure of each predictor, which would be quite an extensive assessment battery. However, in retrospect, it would have been valuable to include a behavioural measure of persecutory beliefs to avoid the possible confound by type of assessment. We have added this aspect accordingly:

Limitations: “Lastly, a higher shared methodological variance of questionnaires compared to computer paradigms might have contributed to differences in the relevance of predictors. Specifically, self-report measures may be more intercorrelated than they are correlated with computer paradigms due to the shared assessment method and its inherent biases. Since this is a common issue even in carefully designed studies using mixed assessment methods^{70,71}, whether and to what extent shared method variance influences the associations between predictors and delusions requires further examination. Including a behavioural

measure of persecutory beliefs in addition to questionnaires may be beneficial to avoid this possible confound.” (p. 20)

Discussion

You mentioned “The latter is partly unsurprising given that some of the psychopathology items 290 overlap with included predictors (e.g., anxiety, depression).” Agree, is not surprising, but is it possible to conduct some sensitivity analysis? For example, did you explore predicting SCL-9 with predictors that remove the general psychopathology, like stress? Perhaps a further analysis to reduce the correlation between general psychopathology as predictor and general psychopathology as outcome.

Response: Thank you for your suggestion. Although clearly several predictors are conceptually close to/overlap with general psychopathology, deciding on which predictors to remove is tricky since the predictors do not unequivocally map onto any of the specific items of the SCL-K-9. Thus, the decision of which predictors to exclude to reduce the overlap would be rather subjective. Further, to compare the explained variance, we would also need to exclude the same predictors from the models of persecutory beliefs and delusion, thus deviating from our study aim.

Nevertheless, your comment sparked another discussion on the interpretation of the findings. On the one hand, and as already stated in the original manuscript, it is not surprising that we were able to predict general psychopathology quite well as some predictors showed overlap with general psychopathology. On the other hand, it is surprising that the prediction worked so much better for general psychopathology than for persecutory beliefs given the predictors were drawn from delusion- and paranoia-specific theoretical models and include both transdiagnostic as well as symptom-specific predictors. We added these aspects to the discussion and hope that this enhances the discussion of our results regarding the specificity of the theoretical models of persecutory beliefs:

Discussion: “The model predicting general psychopathological symptoms performed even better than the delusion-focused models. This strong model performance as such is not entirely surprising, given that some of the psychopathology items overlap with included predictors (e.g., anxiety, depression). Moreover, many of the proposed risk factors, such as negative affect (i.e., anxiety, depression, stress) and emotion regulation difficulties, are known to be transdiagnostically relevant^{34,35}. What was unexpected, however, is that the model prediction worked substantially better for general psychopathology than for persecutory beliefs despite being theoretically tailored to the latter. This finding might support a hierarchical or dimensional model of psychopathology, wherein persecutory beliefs represent one specific expression of a higher-order general psychopathology factor. For instance, the Hierarchical Taxonomy of Psychopathology (HiTOP) proposes unusual beliefs, including persecutory beliefs as one component of a thought disorder spectrum within a psychosis superspectrum³⁶. In this framework, many of the predictors might be acting as indicators of a core liability, whereas persecutory beliefs only emerge when additional conditions are met, thus limiting their predictability compared to the broader construct.” (p. 15)

Reviewer #2 (Remarks to the Author):

Hello – congratulations on an impressive piece of work. I recommend it for publication, with some minor revisions. Since my review is very long, for convenience I have extracted from it, and placed in a numbered list of ‘action points’, all the revisions/clarifications/comments I request of you. (In addition to these, within the review itself there are a couple of other ‘suggestions’ that you don’t have to act on at all – one is a request for more detailed reporting of secondary analyses that would be nice but not necessary and may well be entirely unfeasible anyway; the other is merely a speculative comment that may be of interest and may be relevant to your overall conclusions, if you find it convincing, but which you might not want to pick up on and should equally feel free to just disregard. Since these aren’t ‘action points’, you needn’t feel obliged to respond to them and therefore I they are not included in the numbered list below). The sub-section of the review within which the issue to which each action point below pertains is chiefly discussed, and/or from which the action point has been lifted directly as a quotation, is indicated for reference.

Response: Thank you for your thoughtful and valuable review of our manuscript! We greatly appreciate the time and care you devoted to providing such constructive suggestions. Below we address each of the Action Points you listed, as well as a few additional aspects drawn from the broader review. For an easy overview, we provided the full text of the relevant sections of your general review, from which the action points were derived, in an appendix at the end of this response. We hope this format facilitates easy cross-referencing between the Action Points, our responses, and the changes to the manuscript.

Action Points:

1) From ‘General Summary and Evaluation’ section: “The pre-registration has been followed faithfully, save for one planned analysis which the authors deemed not meaningful to proceed with in light of what they consider a lower-than-expected degree of predicted variance to have been achieved by all included predictors in the random forest analysis. On that point, it is not clear what the exact percentage of variance predicted was ‘expected’ a priori, nor why a failure to meet that threshold of predictive success (whatever it might be) would render the plan to perform iterative exclusion of predictor variables an uninformative analysis not worth proceeding with. Indeed, my understanding is that iteratively excluding uninformative predictors can frequently improve performance of random forest predictions - and in any case the results of such an analysis would have been useful for improving the interpretability of the central findings”. ACTION: The precise technical reasons for which it was decided not to proceed with this planned analysis should be stated and explained, or (alternatively) the analysis should be conducted and reported as planned.

Response: We agree that the rationale for deviating from the preregistered iterative exclusion analysis should be explained more clearly. The idea behind the originally planned analysis was to iteratively exclude predictors from the least to most important (based on SHAP values) in order to explore how many variables are necessary to achieve a reasonable level of predictive accuracy.

However, given the limited gain in predictive accuracy in our model and the substantial computational demands of retraining and evaluating 55+ nested random forest models (each requiring bootstrapping or cross-validation for robust inference), we judged the expected incremental insight to be disproportionately small in relation to the analytic effort involved.

Therefore, we opted to focus on a reduced model with the highest-ranking predictors, which is more interpretable and still consistent with the core aim of identifying variables with the strongest predictive utility.

We have revised the manuscript to clarify this decision and more explicitly acknowledge this limitation of our analytic approach.

Results: “In our preregistration, we had planned to conduct an iterative variable exclusion analysis to determine how many predictors are required to reliably predict persecutory belief scores (www.osf.io/tuayf). However, the full random forest model including all 55 predictors explained only 31% of the variance, indicating relatively modest overall predictive power. In light of this result, we opted not to conduct the preregistered stepwise exclusion of predictors from least to most important, as we deemed the expected additional insights from such an analysis to be limited.

Instead, we conducted an exploratory analysis including only the ten most influential predictors (based on SHAP values). This reduced model yielded a comparable, slightly higher explained variance (35%). While iterative exclusion can, in principle, improve model performance by reducing noise, we judged the analytic effort required for retraining and evaluating more than 50 nested models to be disproportionate to the likely gains in interpretability or predictive accuracy in this particular case. This deviation from the preregistered plan is transparently documented and acknowledged as a limitation.” (pp. 10-11)

Limitations: “Fifth, we deviated from our pre-registered analysis plan due to limited expected additional insight that would be gained from iteratively excluding predictors, given the already limited predictive power of the model, which would be disproportionate to the high analytic effort of building over 50 nested models.” (p. 20)

As requested in the general review (see Appendix), we further provided the same plots for the exploratory analyses as for the primary analyses in the supplementary material (Figures S4 and S5).

Results: “The reduced model, excluding these ten variables, yielded an explained variance of 22%. The scatterplots of the observed and predicted scores and the beeswarm plots displaying the ten variables with the highest SHAP values can be found in the supplement (Figures S4 and S5).” (p. 11)

2) From ‘Review of Scientific Aims and Methods’ section: “It appears that for a cognitive task to qualify as a feasible instrument for inclusion in this study, it had to be quite short in terms of trial-number: I would have expected to see some reinforcement learning component in the study otherwise, and presumably the absence of any such task reflects the large number of trials that would be required to assay computational parameters meaningfully (in contrast to the more economic beads and JtC tasks, from which parameters can be estimated on the basis of a few trials corresponding to a short overall task duration)?” ACTION: Please comment on this briefly in response

Response: Thank you for this thoughtful comment. In terms of reinforcement learning, only “partial reinforcement” was listed in the models, and this was related to maintenance rather than formation and was therefore not included as a predictor. Generally though, and as noted in our response to Reviewer #1, we indeed considered including other paradigms that allow

for the estimation of computational parameters targeting underlying mechanisms. However, such paradigms typically require extensive trial numbers and multiple blocks to yield reliable and interpretable results. In the context of our online study, which imposed strict constraints on task length and participant burden, this was not feasible. We therefore opted for paradigms such as the beads task, which allow for meaningful parameter estimation with a minimal number of trials and within a short timeframe. We acknowledge that the operationalisation restrictions are a limitation of this study:

Limitations: “Third, a few predictors were not included in the study due to the lack of reliable and feasible assessments, our focus on the aetiology of delusions, and substantial conceptual overlap between predictors and persecutory beliefs (see Supplementary Table S1 for details).” (p. 19)

3) ‘*Selection of Causal Models of Delusion Formation for Consideration*’ section: **ACTION:** “...please identify and justify the inclusion of these particular six models, and (if space allows) perhaps also comment on why the systematic review failed to identify some important areas of research (and on whether there are any other lacunae that are likely neglected in the final sample) or whether there were any other theories the authors noticed were likewise not picked up by the review procedure but which they ultimately deemed not as important to include as those six that they did decide it was essential to include, despite the review having failed to identify them.”

Response: Thank you for voicing your concern about the selection of models during our literature search.

We acknowledge that our literature search might not have picked up every published model on delusions, as we did not include models published before 1990 or models focusing on the maintenance of delusions only. However, without exceptions, every model that fit the specified inclusion criteria was included. As with the literature search results, we included every model known to us that fit the inclusion criteria. These were Beck (2008), Denève & Jardri (2016), Feeney et al. (2017), Friston et al. (2016), Greenaway et al. (2019), Morrison (2001), Newman-Taylor et al., (2020), and Preti & Cella (2010). Some of these models were not indexed in the databases searched (e.g., Beck, 2008). For others, we could not find a systematic explanation why they did not show up in the literature search. Despite our efforts, we thus cannot rule out the possibility that some models did not appear in our literature search. We added more detail to the description of the literature search in the methods, acknowledge that this is a limitation, and identified the models that we added from our knowledge in the description of the literature search flow chart in the supplement:

Methods: “This search resulted in 51 publications that fit the inclusion criteria (see Supplementary Figure S1 for details). Two additional models were published after the data collection for the present study (i.e., ^{17,18}).” (p. 5)

Limitations: “Fourth, despite our efforts, we cannot entirely rule out the possibility that some existing models of delusion formation that would have met our inclusion criteria were not identified in our literature search.” (p. 20)

Supplementary Figure S1: “Eight models that did not appear in the literature search but were known to the authors and fit the inclusion criteria were included (i.e., refs. ¹⁻⁸)”

References in the Supplement:

1. Beck, A. T., Rector, N. A., Stolar, N. & Grant, P. *Schizophrenia: Cognitive Theory, Research, and Therapy*. (The Guilford Press, New York, 2009).
2. Denève, S. & Jardri, R. Circular inference: mistaken belief, misplaced trust. *Curr. Opin. Behav. Sci.* **11**, 40–48 (2016).
3. Feeney, E. J., Groman, S. M., Taylor, J. R. & Corlett, P. R. Explaining delusions: Reducing uncertainty through basic and computational neuroscience. *Schizophr. Bull.* **43**, 263–272 (2017).
4. Friston, K., Brown, H. R., Siemerikus, J. & Stephan, K. E. The dysconnection hypothesis. *Schizophr. Res.* **176**, 83–94 (2016).
5. Greenaway, K. H., Haslam, S. A. & Bingley, W. Are “they” out to get me? A social identity model of paranoia. *Group Process. Intergroup Relat.* **22**, 984–1001 (2019).
6. Morrison, A. P. The interpretation of intrusions in psychosis: An integrative cognitive approach to hallucinations and delusions. *Behav. Cogn. Psychother.* **29**, 257–276 (2001).
7. Newman-Taylor, K. *et al.* Cognitive mechanisms in cannabis-related paranoia; Initial testing and model proposal. *Psychosis* **12**, 314–327 (2020).
8. Preti, A. & Cella, M. Paranoid thinking as a heuristic: Paranoid thinking as a heuristic. *Early Interv. Psychiatry* **4**, 263–266 (2010).“

4) ‘Conceptual Overlap Between Predictors and Outcome’: “The exclusion of schizotypy, on the grounds that it shares too much conceptual overlap with the target outcome (i.e. paranoid thoughts), seems strange to me. While I am highly sympathetic to the logic itself, by that very logic it seems troubling that many of the questionnaires that were included – and, importantly, that ended up ranking high on overall predictive importance – seem to share at least as much conceptual overlap, if not more, with that target outcome than does schizotypy.” ACTION: this whole issue is by far the most serious concern I have about the paper, and I would ask that you engage with it not only in your response but also (since I think it likely other readers will be troubled by this too, if it is not addressed head-on) that you discuss it in the manuscript itself. This discussion should either respond to the concerns satisfactorily, or (equally acceptably and perhaps more feasibly) simply acknowledge the issue as a limitation and consider what it might mean for the overall conclusions that can be drawn from the work (e.g. the greater significance, given their lack of conceptual overlap with the outcome and thus their relative disadvantage compared to most other top-performing predictors, of factors like ‘aberrant salience’ and ‘hallucinations’). I feel strongly that this issue needs to be given discursive treatment in the text, but I do additionally suggest (within the relevant section of the review) a couple of ways in which more complete reporting of results could augment – though certainly I don’t think it could replace – this verbal discussion. However, space and feasibility may not permit the fuller reporting of results I suggest, and in any case the issue needs to be tackled head-on in discussion, which need not refer to any results not already reported unless you feel it useful to do so. So long as you engage with the issue seriously in the text (either to recapitulate the points I have raised and acknowledge them – or else to introduce and somehow refute them and convincingly show why, after all, schizotypy alone ought to have been excluded) then I will be perfectly satisfied on this point.

Response: Thank you for raising this issue and for your suggestions on how to deal with it.

Your point taps into a general problem that the target outcomes are more or less close to the predictors included to predict them. While the general rule was to include all factors mentioned in the models we made a few exceptions. Reasons for exclusion were a) practicability (length of a task/questionnaire), b) no reliable assessment of variable possible, c) proposed primary role in the maintenance of delusional beliefs rather than the formation, and d) strong conceptual overlap to persecutory beliefs (i.e. if the target variable is the same as or is an essential part of the predictor rendering the prediction tautological). To increase transparency, these criteria were named more explicitly in the manuscript:

Methods: “Although the general principle was to include all predictors outlined in the models, a few exclusion criteria were set. Predictors were excluded for the following reasons: a) practical constraints, such as the length of the task or questionnaire, b) the absence of a reliable method for assessing the variable, c) primary involvement in the maintenance of delusional beliefs rather than the formation, and d) substantial conceptual overlap with persecutory beliefs (i.e., in cases where the target variable was either identical to or a core component of the predictor, rendering the prediction tautological). A list of excluded predictors is provided in Supplementary Table S1.” (p. 5)

Schizotypy is commonly defined as a cognitive-perceptual aberration (including ideas of reference and paranoia) as well as interpersonal deficits and disorganisation and represents a diagnostic category. Thus, persecutory beliefs are, by definition, a part of schizotypy. Therefore, we considered that including it as a predictor would be tautological. In contrast, other predictors (e.g., trust, anticipation of threat) can be considered conceptually close but only overlap with aspects of persecutory beliefs and do not fulfil this criterion. However, we agree that other criteria and boundaries for exclusions may also have been justifiable. We elaborated on this limitation accordingly:

Limitations: “Third, a few predictors were not included in the study due to the lack of reliable and feasible assessments, our focus on the aetiology of delusions, and substantial conceptual overlap of predictors with persecutory beliefs (see Supplementary Table S1 for details). Although maintenance mechanisms were not included in this study, they may already play a role in the aetiology of persecutory beliefs by reinforcing and exacerbating threat beliefs once they occur. For instance, recent studies highlighted the role of safety behaviours in both the development and maintenance of persecutory beliefs (e.g., Freeman & Loe, 2023). We acknowledge that other criteria and boundaries for excluding predictors may also have been justifiable. Further, the decision of when a factor postulated in a model is conceptually too close to persecutory beliefs to be included remains somewhat subjective.” (p. 19)

5) From ‘Conceptual Overlap Between Predictors and Outcome’ section: ACTION: please describe the distribution (e.g. descriptive statistics and/or a figure illustrating this) of the temporal delay between screening and main experiment sessions, for each group. Comment on whether you think this temporal delay is likely to have been a concern or not

Response: We computed the temporal delays from starting the screening to starting the main survey. The median delay is in the range of hours rather than days and is approximately equal across groups, except for a descriptively longer delay in the moderate quota. Such short delays are unlikely to be a problem for the variance in persecutory beliefs and the prediction of the model.

We added the median delays for the whole sample and the quotas in the methods section and briefly discussed the possible effect of the delays in the discussion:

Methods: “The median (Mdn) delay between starting the screening and the main survey was 9.29 hours (median absolute deviation [MAD] = 11.71). The median delay was longer in the moderate quota (Mdn = 16.17, MAD = 15.16) than in the other three quotas (average: Mdn = 7.34, MAD = 8.62; elevated: Mdn = 9.39, MAD = 11.58; severe/very severe: Mdn = 8.51, MAD = 10.74).” (p. 6)

Limitations: “Further, predictors were assessed with some delay due to the screening procedure. The delay between screening and main survey was less than ten hours on average and was approximately equal across the quotas, with descriptively longer delays in the moderate quota. Such short delays are unlikely to have influenced the variance of persecutory beliefs and the resulting model predictions.” (p. 18)

6) *‘Improving the Introduction and Explanation of a Novel ML Method’*: **ACTION:** “At present, the cited references do not constitute anything approaching a sufficient reading list to supply the relevant background (they are neither numerous enough, introductory enough, or specifically germane enough to the point being made by the sentence within which they are cited) – many terms that could usefully be explicated in the main text (e.g. ‘nested cross-validation’, ‘inner folds/outer folds’) are left undefined, and the average reader will have to be motivated to independently undertake a great deal of investigative reading elsewhere, if they are to be in a position to infer the reasons for each of a number of very cursorily brushed over methodological choices in the details of this machine learning approach’s implementation. Though I am satisfied that the methodological choices are suitable for the purposes to which they are intended (as a result of my having undertaken to do this additional research in the service of writing this review) there is a missed opportunity to make the paper’s conclusions more intelligible, which could be rectified were its novel machine learning methodology more engagingly and transparently presented. I believe this would greatly improve the paper’s impact (in every sense of that word). The novelty of the application of random forest machine learning prediction methods to delusion research is a strength of the work - but only to the extent that the majority of its probable readership is able to meaningfully engage with and appreciate the logic underlying it.” I mention this general concern at several points throughout the review, and while it may sound like I am asking for a substantial rewrite, in fact I believe quite a little effort in the direction I have suggested would go a long way in making things easier and clearer for a reader new to these methods.

Response: Thank you for going to such great lengths to review our manuscript!

To better explain the machine learning approach, we supplemented the methods section with definitions of important terms and added a less technical explanation of how random forests and nested cross-validation work. Further, we highlighted why machine learning is a well-suited approach for the aims of study in the introduction and methods section. Lastly, we provide published articles for an easy introduction to machine learning and random forests for interested readers (i.e., Cutler et al., 2012, doi:10.1007/978-1-4419-9326-7_5; Breiman, 2001, doi:10.1023/A:1010933404324, and Turgeon & Lanovaz, 2020, doi:10.1007/s40614-020-00270-y):

Introduction: “In random forests several decision trees are created and predictions (e.g., levels of persecutory beliefs) are calculated by averaging the individual predictions of all decision trees.” (p. 4)

Statistical Analyses: “Random forests are an ensemble learning method that calculate a multitude of decision trees, each on a random subsample of a training dataset^{13,14,30}. To generate predictions (e.g., levels of persecutory beliefs), individual decision tree outputs are averaged for metric outcomes. In the case of categorical outcomes, the random forest prediction constitutes the majority vote (e.g., high levels vs. low levels of persecutory beliefs) across all decision trees. We built our random forests in multiple steps. First, we used nested cross-validation with ten outer and ten inner folds (i.e., random splits of the data) to tune the hyperparameters (e.g., the number of trees in the forest) of the random forests and evaluate their performance. In each outer split, the data is first divided into an outer fold training and testing dataset. Then, the outer fold training dataset is partitioned further into ten inner folds consisting of a training and validation dataset each. Various random forests with different hyperparameter configurations are trained on the inner training datasets and evaluated using the corresponding inner validation dataset. This approach enabled us to identify the best hyperparameter configuration that yielded the highest average performance (e.g., R^2) across the inner folds. Once the optimal hyperparameter configuration is identified in the inner folds, a random forest is calculated on the entire outer training dataset using this optimal hyperparameter configuration. Then the trained random forest is applied to the outer fold test dataset to evaluate its performance. This procedure is repeated across all outer folds and is used to reduce the risk of overfitting.

We tested for all possible hyperparameter configurations during hyperparameter tuning (i.e., grid search) using the hyperparameter space (i.e., possible values of the hyperparameters) displayed in **Supplementary Table S4**. Subsequently, the best hyperparameter configuration (i.e., with the highest explained variance [R^2] across the validation datasets in the inner fold) was used to train the random forests on the respective outer fold training data. These random forests were then applied to the outer fold testing data to evaluate the explained variance of the prediction model (Q1). To test which variables best predict persecutory beliefs (i.e., R-GPTS persecution; Q2), we calculated Shapley Additive exPlanations (SHAP)³¹ during the nested cross-validation procedure. These SHAP values represent the individual contribution of each variable to the prediction (e.g., the contribution of one’s anxiety score towards the random forest’s prediction of the respective person having high levels of persecutory beliefs) and are calculated taking all other predictors and their presumably complex interactions into account.” (pp. 8-9)

7) ‘Screening Procedure and Interpretability of High Scores Absent Concomitant Diagnostic Prevalences’: “However, there is a striking disparity between the psychometric cut-off corresponding to the “likely” presence of clinical delusions of persecution (11) and the total absence of self-reported diagnosed psychotic illness within the ‘moderately severe’ group (scoring 11-18). Moreover, there were only two psychotic illness diagnoses among the equally large ‘severe’ group in the sample, - which is even more puzzling given that the lower threshold for inclusion in this latter group (a score of 18) has been previously shown to constitute a ‘cut-off [that] is unlikely to identify incorrectly an individual as having a persecutory delusion when they do not.” ACTION: “A couple of sentences further consideration of the issue would do wonders to reassure the reader about the relative insignificance of this potential issue or (equally acceptably) to highlight it as an actual, substantive limitation.” Within the relevant

section I suggest a couple of possibilities for the direction your response to this point might take, however of course you may take another tack entirely – all that matters is that you make some attempt to explain the anomaly in question and briefly explore its implications (or alternatively, that you demonstrate my concern to be entirely misguided,).

Response: Following your suggestion, we evaluated the frequency of self-reported diagnoses as well as treatment received per quota and added an overview in the supplementary material (Table S2). Participants from higher quotas reported having a diagnosis of depression and/or anxiety disorders more often than participants from lower quotas, suggesting higher psychological distress. Moreover, we can speculate that perhaps individuals underreported psychosis diagnoses due to the still prevailing stigma regarding schizophrenia. Nevertheless, whether the results can be replicated in samples with a higher reported number of psychosis diagnoses remains to be determined. We have addressed this issue in the manuscript accordingly:

Results: “An overview of the reported mental health diagnoses per quota is provided in Supplementary Table S2.” (p. 10)

Limitations: “Moreover, persecutory beliefs are common in psychotic disorders and one might have therefore expected a higher-than-average (0.30-0.66%)⁶⁸ prevalence of psychosis despite our sampling strategy not being designed to specifically recruit individuals with a diagnosis. However, only two participants (0.6%) reported a diagnosis of a psychosis-spectrum disorder. It is possible that such diagnoses were underreported due to stigma associated with schizophrenia spectrum disorders^{68,69} or that our sampling strategy was biased towards individuals without a diagnosis. Importantly, psychological distress was not generally low in our sample: participants in higher quotas of persecutory beliefs reported depression and anxiety diagnoses as well as currently receiving mental health treatment more frequently than those in lower quotas. Nonetheless, whether the current findings replicate in samples with a higher prevalence of diagnoses of schizophrenia-spectrum disorders remains an open question that should be addressed in future studies.” (p. 19)

This puzzling disparity between the prevalence of psychotic disorders that would be expected in such high scorers, and the very low rates of same actually reported within these groups, warrants reflection. If the scale were functioning here (in the present study) in roughly the same way as it did in the studies responsible for validating it and establishing its clinical cut-off points (i.e. if the scale in the present context were working simply ‘as it should’) then that would suggest a great number of participants within the upper two quadrants of severity can, with a high degree of confidence, be identified as individuals who very likely are suffering from undiagnosed and untreated clinical psychosis. If this is deemed the correct interpretation - was there any provision made in the ethics for contacting these individuals and signposting them to appropriate clinical attention/care?

Response: To ensure anonymity, we did not reach out to individuals with high scores in persecutory beliefs. However, we provided the contact details of a UK telephone counselling service (Mental Health First Aid: (020) 7250 8062) in the study information, which could be contacted in case of distress. Further, participants were able to contact the first author via E-Mail or anonymously via their Prolific dashboard.

Comment: Failure to Predict Paranoia, or Failure to Measure it?:

The persecutory subscale of the R-GPTS alone may be less suitable, as an instrument for the measurement of specifically persecutory delusions, when used in isolation as compared to when used in combination with the 'ideas of reference' subscale of that same instrument - or even perhaps compared to just the 'ideas of reference' subscale alone, which (in clinically high-risk persons, at any rate) seems to more purely single out persecutory delusions per se than does the so-called 'persecutory' subscale (which by contrast seems to better capture general social functioning deficits) (Williams, Trevor F., et al. "The reliability and validity of the revised Green et al. paranoid thoughts scale in individuals at clinical high-risk for psychosis." Acta Psychiatrica Scandinavica 147.6 (2023): 623-633.).

In this regard, the better performance achieved by the random forest for predicting the combined score of these two R-GPTS subscales (compared to its performance for predicting scores on the persecutory subscale alone) may be interpreted as indicative of the closer relationship between the target construct for which all predictors were selected on the basis of their aetiological contribution to under current models (i.e. with the principal outcome of paranoid delusions specifically and a measurement instrument that was here used to assay another more general construct (i.e. the secondary outcome measure of 'paranoia plus reference'). This secondary outcome measure may, in actual fact, have more psychometric relevance to the primary target construct that the study first and foremost intended to predict (and, therefore, that it may have succeeded in doing so, more successfully than the authors perhaps realised).

To put this another way, it may be that the poorer prediction of "paranoia" (as operationalised by the persecutory subscale of the R-GPTS in this study) compared to the prediction of scores on another instrument, that was used as a secondary outcome measure here (i.e. the combined R-GPTS persecutory and reference subscales), reflects not poor prediction of paranoia by the causal factors theorized to cause it, but better measurement of paranoia by the instrument that was better predicted in the present study than was the instrument used to measure paranoia itself. If so, this secondary measure's better-than-anticipated prediction by the curated aetiological factors in question is suggestive not of those factors' failure to specifically capture aetiological processes important in the primary outcome construct of interest (paranoia), but rather of the better suitability of the combined instrument (compared to the persecutory subscale alone) for measuring paranoia. This seems to be in the authors' favour, and might be worth considering in their evaluation of their model's success – an evaluation which is admirably thoughtful and measured, but may be permissibly rendered a little bit more optimistic in light of what I suggest in the present paragraph, if the authors consider the above suggestions persuasive and if they have a mind to revise that section to include the cited reference that supports them.

Response: Thank you for pointing out this interesting study and sharing your thoughts on its implications for our findings. As suggested, we added a brief discussion of this study and possible interpretations regarding our findings:

Discussion: "Compared to persecutory beliefs, the ML models explained more variance in the combined R-GPTS persecution and reference subscales and in the more broadly defined delusions in general. This could result from broader constructs being more easily explained compared to more narrowly defined, specific symptoms. Alternatively, it could be due to a better alignment of the combined R-GPTS scale to the target construct. Specifically, a recent

study found the persecution subscale to be more closely related to clinician-rated functional deficits, whereas the reference subscale was more closely related to clinician-rated persecutory ideas³³. Accordingly, it has been recommended that both subscales should be used when examining the relationship between persecutory beliefs and its putative predictors or related factors³³. Considering this, the relatively poorer performance of the model predicting persecutory beliefs alone may not reflect a failure of the proposed predictors to explain persecutory beliefs, but rather a better suitability of the combined scale to fully capture the construct.” (p. 14)

Two further changes were made to increase the transparency of our manuscript:

We added more detail on the software used for data preparation and plotting: “R version 4.4.3 was used for calculating sumscores and sample characteristics, and for preparing the prediction plots (Figure 2). The *fmincon* function in MATLAB (version R2022b) was used for modelling the Bayesian inference variables (see Table 1, Beads estimation task).” (p. 8)

We pointed out that the means and standard deviations of all predictors can be found in the Supplement in the Results section: “Means and standard deviations for all predictor variables are provided in Supplementary Table S3.” (p. 10)

We are confident that these improvements have strengthened the quality and rigour of our manuscript and thank you for your time and support.

Kind regards,

Saskia Denecke

(on behalf of all authors)

Appendix

Reviewer #2

REVIEW:

General Summary and Evaluation:

This work is a novel and timely attempt to objectively compare a systematically curated set of the various causal factors that are implicated by current aetiological models of persecutory delusions (and that are, additionally, amenable to operationalization within an online population study) with respect to their importance as predictors of subjects' previously-obtained scores on a well-validated self-report measure of persecutory ideation (the R-GPTS questionnaire's persecutory subscale). In an initial online screening, members of the general population were administered this questionnaire and invitations to participate in the main online study were

selectively issued in order to obtain a stratified sample that ensured equivalent representation of subjects within all levels of paranoia severity. This allowed random forest machine learning methods to be applied to the task of predicting paranoia severity on the basis of the included causal factors. The recruitment, sampling and testing methods are appropriate and their details have been judiciously decided with respect to the intrinsic trade-offs involved (trade-offs which the authors are cognizant of and which they thoughtfully grapple with in interpreting their findings). The discussion section is an excellent piece of scientific writing: its claims are clearly presented and cautiously evaluated, and the argumentation is convincing. However, I would like to see it expanded somewhat to address some concerns I have about the conceptual overlap that seems to exist between the target construct at issue and many of the included predictors thereof (particularly the most dominant predictors as per the random forest analysis's estimation of their importance). I also have some concerns about the interpretation of such high scores on R-GPTS measures of paranoid ideation as would typically be concomitant with clinical cut-offs for psychotic delusional illness, within an online quota sample in fact reporting very low prevalence of such disorders. Conceptual overlap of predictors and outcome, and the interpretability of scales intending to measure paranoia in this screened online sample, are issues that would benefit from some further consideration within the discussion section of the manuscript.

The random forest machine learning methodology employed to predict persecutory delusions and to estimate the importance of each of the included causal factors for generating these predictions, is generally well-suited for the intended purpose, and the particulars of its implementation seem well-chosen. Unfortunately, these analytic methods are not fully enough described and explained for this favourable conclusion to be apparent, without the reader first undertaking considerable additional independent reading to familiarize themselves with these approaches. Only by having done so, can readers who are not already familiar with random forest machine learning (such as myself, and presumably the majority of the paper's intended audience) understand what was done, and infer why it was done, in this study's analysis. This is a pity, as the novelty of the method's application (to a problem the authors rightly identify as hampering progress in the field of delusion research as a whole and which their methodology was deliberately chosen to address neutrally and comprehensively with respect to all factions within that field) is a strength of the work. This strength ought to render the work more (not less) widely read and engaged with by delusion researchers – however, currently the conceptual basis of the analytic methods used are very underexplained, and the reach and readability of the work is likely to be reduced due to a missed opportunity to more didactically introduce the conceptual basis of the analytic methods (in recognition of its novelty and in anticipation of the epistemic needs of the large number of likely interested readers).

The pre-registration has been followed faithfully, save for one planned analysis which the authors deemed not meaningful to proceed with in light of what they consider a lower-than-expected degree of predicted variance to have been achieved by all included predictors in the random forest analysis. On that point, it is not clear what the exact percentage of variance predicted was 'expected' a priori, nor why a failure to meet that threshold of predictive success (whatever it might be) would render the plan to perform iterative exclusion of predictor variables an uninformative analysis not worth proceeding with. Indeed, my understanding is that iteratively excluding uninformative predictors can frequently improve performance of random forest predictions - and in any case the results of such an analysis would have been useful for improving the interpretability of the central findings. The precise technical reasons for which it was decided not to proceed with this planned analysis should be stated and explained, or (alternatively) the analysis should be conducted and reported as planned.

As a suggestion: if space permits and if the authors consider this feasible, it may be worth extending the supplementary material in order to report for all secondary and exploratory analyses (e.g. for the analysis omitting the first ten most important predictors) the same plots and level of detail as was afforded to the primary analysis, not just for completeness' sake but because these would be interesting in their own right (although again, I understand that it may be impracticable to implement this, so I suggest it merely as an option that for the authors to consider). On that note (and of some relevance to the issue of 'lower than expected' predictive success touched on above) in a later section of the review, I speculatively suggest (citing recent evidence) that, after all, the predictive success of the model – i.e. its predictive success evaluated with reference to the target construct of interest: paranoid delusions (as distinct from the instrument used to assay that construct) – may actually be less disappointing than the authors estimate it as having been. However, this idea is not in any sense proffered as something the authors need to include or engage with in their manuscript: it's not a suggested revision at all, merely a somewhat speculative pointer towards what could, if the authors judge that it would be beneficial to reframe things in this way, allow for a rather more optimistic evaluation of the degree of success with which they have managed to predict paranoia in the general population.

To summarise, I recommend this work for publication. At the same time, I suggest some minor revisions to the manuscript that I believe would improve its interpretability, and increase the reach and impact of the valuable work it reports. Aside from the general suggestion that the methodology be introduced and explained more fully, these suggestions mostly centre around the need to discuss certain outstanding conceptual issues (the most important of these being the 'conceptual overlap' issue detailed below, which I found particularly troubling), and in a few cases I request that the authors provide information about some specific detail if possible. My suggestion that the random forest methodology be introduced more fulsomely may be quite accurately seen to merely reflect my own unfamiliarity with the approach (an approach which I should clarify that I have, as a prerequisite to reviewing this work, since familiarised myself with through careful background reading, and which I am as a result now confident I understand adequately – although not of course expertly). However, my position as a novice in this regard is hardly likely to be unique to (nor even atypical of) the readership of the paper, and indeed many psychological theorists of psychosis who do not regularly use computational methods of any sort in their own research may be even less well-positioned than myself to get by without further assistance on the part of the authors. I therefore hope that as a non-expert in machine learning, my advice (reflecting as it undoubtedly does my own ignorance) will be all the more useful for coming from the same epistemic position from which the paper will be approached by most of its readers. This is an excellent piece of work, and I enjoyed reading it and thinking about it. The minor revisions I suggest are intended to help the authors present it in a manner that does it justice, and that pre-emptively addresses what I envisage would be the most likely questions or concerns about it among a readership that I fully hope and expect will be a large and appreciative one.

Review of Scientific Aims and Methods:

The authors identify a problem for current research into the causes of paranoid delusion formation, which their study aims to address: despite the proliferation of aetiological models which continues apace, the field has failed to make definitive progress on explaining delusion formation anywhere concomitant with the calibre of the work being done in service of that agenda. That is, the question of what causes paranoid delusions – which everyone agrees has

no unitary answer – ought to at least be answerable in terms of which of the many causal factors involved are major contributors (and which are relatively minor, or central only to certain paranoid presentations). No such ranking has been agreed upon because the development of causal models lacks constraint: there has been little in the way of systematic, impartial and scoping quantitative investigation into the relative predictive importance of the explanatory variables that are each central to one or another of the various models of paranoia's aetiology. As a result, the present Cambrian explosion of theoretical frameworks for understanding delusions formation lacks a concrete, externally-grounded basis for adjudicating between the vast array of aetiological models, in terms of which of these are the most likely to in fact explain a greater part of the overall variance in the observed phenomenon. This makes it difficult to prioritise the most promising avenues for future research, since to do so would require some viewpoint-agnostic, reliable metric for estimating the explanatory promise afforded by any given line of enquiry relative to its alternatives. The authors rightly note that there is not much direct competition between current theories of delusion formation – since each posits the central importance of some aetiological factor(s) that could - and (in large part at least) very likely do – causally explain some portion of the total variance, in who does and does not develop delusional beliefs, at a population level. The causes are complex and multifaceted and interactive, so adherents to one model have little independent reason to redirect their attention to another, absent some objective and universally-acceptable metric for gauging which specific factors appear to be particularly important in explaining variance in delusion formation within the population of interest.

The authors' aim in this paper is to provide such an impartial analysis: quantitatively estimating the relative importance of causal factors implicated by current theories, with a view to guiding future research towards the most promising avenues of enquiry. This aim is a laudable one, and the attempt to offer a relatively 'neutral' metric for evaluating the merits of alternative theories using first a systematic review of the literature (to select candidate causes) and then machine learning (to rank their predictive importance) is well conceived. The population for which the authors thus attempt to predict variance paranoia severity is the general population, while the causal models included are garnered in the main from theories of how clinical paranoid delusions form (e.g. schizophrenia). Thus, the authors tacitly rely on a continuum approach to conceptualizing psychotic experience, in order to draw inferences from (A) "the relative importance of these factors for predicting paranoia severity in a non-clinical population", about (B) "the relative promise of the causal theories that invoke those factors to explain clinical paranoia". The contingent of clinically-oriented researchers who reject the notion of a meaningful continuum between the processes underlying non-clinical persecutory ideation and those underlying clinical delusions proper (or who insist only upon such a categorical distinction only when it comes to delusions in some particular subset of patients e.g. delusions in schizophreniform illnesses) will understandably find this work less than relevant to understanding anything about a clinical phenomenon that they deny exists on a continuum with the kind of persecutory ideation measurable by validated questionnaires designed for general population use. That said, the causes of persecutory ideation in the general population are undoubtedly worthy of investigation in its own right - even if they are categorically different in kind from all (or some) clinical forms of delusion. (Indeed, understanding the causes of delusions in different populations and examining the degree of overlap or divergence between them is information that would very relevantly contribute to resolving debates around continuum vs categorical approaches). Moreover, the majority of researchers do accept the utility of a continuum framework for thinking about delusional beliefs: for readers not opposed to the continuum approach in principle, the application of it here will

be judged an acceptable one, as the study avoids some common pitfalls associated with the continuum approach by taking a number of careful methodological steps (e.g. attention checks peppered throughout the main experiment, and the choice of a questionnaire instrument which has been well validated with respect to clinical cut-offs in previous patient studies as a primary outcome measure).

At all stages (from selecting a set of candidate causal models, to determining which ones to include, to evaluating their predictive importance quantitatively) there is an admirable (and generally successful) effort to maintain objectivity (as befits the study's overall purpose and the problem it sets out to tackle) evident in their choice of methods:

(1) an independent literature review of current theories of delusion formation, supplemented with six unidentified additional models which the authors were personally aware of and that they judged ought to be included, despite the literature review having failed to identify them using systematic methods

(2) a selection process by which this pool of models was winnowed down to only those for which the construct invoked as aetiologically important could be measured in an online study by means of a well-validated instrument: very often a questionnaire, though not uncommonly a cognitive task - and occasionally a computational model of cognitive task performance. It appears that for a cognitive task to qualify as a feasible instrument for inclusion in this study, it had to be quite short in terms of trial-number: I would have expected to see some reinforcement learning component in the study otherwise, and presumably the absence of any such task reflects the large number of trials that would be required to assay computational parameters meaningfully (in contrast to the more economic beads and JtC tasks, from which parameters can be estimated on the basis of a few trials corresponding to a short overall task duration)? In one instance only (which seems not particularly uniquely deserving of its singular fate) a model was excluded because it was deemed to conceptually overlap to a significant degree with the target construct i.e. delusional thoughts, especially paranoid ones.

(3) a random-forest machine learning analysis to determine how important were each of these measures, in terms of contributing usefully to predicting previously-collected paranoia severity scores, within a paranoid-thoughts-stratified general population sample (selected by quota for comparable representation within each of four paranoia severity brackets, from a broader participant pool screened on the R-GPTS paranoia subscale questionnaire for paranoia severity). Predictors were combined with one another stochastically, across a large number of individual decision tree algorithms (each with access to a different with-replacement subset and tasked with predicting R-GPTS paranoia (and, in subsequent analyses, R-GPTS paranoia + reference ideas; delusional ideation tout court as measured by the PDI-21; and finally general psychopathological symptomatology)).

Selection of Causal Models of Delusion Formation for Consideration:

The authors' aim to provide a relatively value-independent, neutral assay of current causal models of paranoia is a laudable one. To curate the set of candidate causal models that may be considered for inclusion, they surveyed the field as a whole by conducting a systematic literature review of current theories of delusion formation. This set of candidates was then supplemented with six unidentified additional models which the authors were personally aware of and which they believed (I have no doubt rightly) ought to have been included on the basis of their merits and importance, despite the literature review having somehow failed to identify them within the purview of its own systematic methods and pre-defined scope.

I have minor concerns about the 'value-laden' decision to include an additional six causal models that didn't crop up in the systematic review but which the authors knew ought to be included (I myself could have thought of other worthy additions – though I refrain from listing them here since the point is emphatically not that 'twelve rather than six' ought to have been appended to the systematically curated pool of candidates derived from the review!). Although I trust that the lucky six were genuinely (even if not uniquely) deserving of their inclusion post-hoc, I would be curious to know which six models these were - this information is currently not reported anywhere, and the manuscript should be revised to include it for transparency. However, no matter how strongly I would (once I knew which they were) tend to agree on the importance of the six models in question, that extra-procedural decision does rather open the authors up to challenge from any reader who finds a model to which they themselves are particularly attached to have been left out – “since the systematic review wasn't binding anyway, whither my pet model?”. This somewhat undermines the commendable objectivity of the method of theory selection in the first place, which was to my mind one of the strengths of the research design - given that the study's intention is after all to provide some viewpoint-independent summary of the relative predictive importance of the various theories that comprise the current state of play within delusion research. I am tempted to start kvetching about the specific models I wish the authors had included, but what I ultimately wish (and what I think would represent the only clean solution to the problem of candidate model selection) is rather that they had included nothing but what the systematic review identified, or alternatively (had they noticed that the methodology of the systematic review was inadequate to capturing the state of the field, given that some work they were aware ought to be included was not identified within it) that they had come up with an expanded set of search criteria to ensure a more thorough but still unprejudiced sampling of the field as a whole, and conducted a literature review in accordance with those updated criteria. This is a quibble for which the best correction, at this juncture, would be as follows: please identify and justify the inclusion of these particular six models, and (if space allows) perhaps also comment on why the systematic review failed to identify some important areas of research (and on whether there are any other lacunae that are likely neglected in the final sample) or whether there were any other theories the authors noticed were likewise not picked up by the review procedure but which they ultimately deemed not as important to include as those six that they did decide it was essential to include, despite the review having failed to identify them.

Conceptual Overlap Between Predictors and Outcome:

The exclusion of schizotypy, on the grounds that it shares too much conceptual overlap with the target outcome (i.e. paranoid thoughts), seems strange to me. While I am highly sympathetic to the logic itself, by that very logic it seems troubling that many of the questionnaires that were included – and, importantly, that ended up ranking high on overall predictive importance – seem to share at least as much conceptual overlap, if not more, with that target outcome than does schizotypy. In their very content, paranoid delusions exhibit a manifest and intrinsic correspondence with the tendency to endorse statements indicating that one is subject to “ostracism”, that one “anticipates threat”, that one distrusts others, and that one generally has negative beliefs about other persons. The commonplace (indeed, even typical) paranoid delusion that I am ostracised and threatened by a conspiracy of my fellows, merely because it is present, will by virtue of its very nature be reflected in my endorsements of such questionnaire items. It will therefore be strongly predicted by questionnaires that score such endorsements highly – but this does not at all imply that there is necessarily any aetiological type of relationship between (for example) the actual state of “ostracism” (as the questionnaire, and its interpretation in the paper, conceives of it) and my state of “paranoia”

(which in fact is directly constituted by my falsely believing that I am ostracised). Nor is there any necessary aetiological relationship between “threat anticipation” and a prototypical presentation of paranoia which constitutively involves anticipated threat (and will therefore be associated with high scores on questionnaires measuring same). The fact that my delusional state happens to revolve around the notion that I am besieged by immanent threat (as paranoid delusions very generally do) does not necessarily result from any pre-existing tendency on my part towards elevated “threat anticipation” more generally – and in an experimental design that measures my paranoia temporally prior to the stage at which it then obtains my measures on all possible predictors of paranoia (it is not clear from the paper how much time before - though it would be good to know this!), then there is no reason to even suspect that my elevated threat anticipation temporally preceded, never mind caused, a paranoid delusion which essentially constitutes an elevated anticipation of threat, in and of itself. To make the point another way: people do not tend to get delusional beliefs that there is something wrong with their dopamine system, even though dopaminergic dysregulation happens to be a true cause of delusions in general; the infidelity of one’s romantic partner (while perhaps in some general sense an example of what could be termed a ‘stressor’ that marginally elevates one’s risk of psychopathology) is not in any meaningful sense a cause of delusions, but it is a common theme in their content and a predictive relationship would be entirely expected to exist between “severity of delusions (measured yesterday)” and “scores on questionnaires tapping into experiences of being cheated on and preoccupation with same (administered today)” as a result of that semantic overlap alone. By the same token, it so happens that paranoid delusions tend almost definitionally to have as their content concerns about threat anticipation, the conviction that one is ostracised, the experience of distrust in one’s fellows etc. But the conceptual overlap between the typical self-reported beliefs constitutive of the outcome variable (paranoia) and the putative causal factors thereof (pre-existing trait mistrust, social ostracism proper, etc - factors that, since maintenance factors are explicitly outside the scope of this study and insofar as possible excluded from it, presumably had to temporally predate the outcome’s emergence in order to count as candidate causes of that outcome) by itself entails that questionnaires probing such experiences administered in the main experiment will have some predictive relationship with paranoid delusions assessed during screening. This would hold true even were it (hypothetically) the case that there was zero causal relationship whatsoever between true ostracism/antecedent tendencies towards threat anticipation or mistrust/etc, and the formation of delusionally paranoid states. It is likely that if I surveyed people with a scale whose items asked about their experiences of victimisation by conspiratorial agents, there would be a predictive relationship between people’s scores and the severity of their paranoid delusional ideation – and that would be a genuine predictive relationship – however it would not constitute evidence of any aetiological relationship between “actual conspiratorial victimisation” (or for that matter, between “pre-existing trait levels of propensity towards belief that one is conspired against”) and “the delusional states which commonly have that narrative scenario as their content”.

Similarly, given what we know about the typical phenomenological form (as distinct from the content) of delusions, it is not surprising that items like ‘suspiciousness (i.e. the content of the belief itself) feels uncontrollable’ (from the scale measuring ‘negative beliefs about mistrust’), and items like ‘I tend to get very entangled in my own thoughts’ (from the scale measuring ‘cognitive fusion’), are important predictors of the presence of paranoid delusional states: these instruments pick up on experiences that are essential and intrinsic to delusional thoughts’ phenomenology.

It is noticeable that the causal models which crop up as the most important predictors are, in a high proportion of cases, those that are measured with an instrument which directly probes some or other example of the most core and typical aspects of the target outcome to be predicted (i.e. of paranoid delusions) – suggesting that the problem of conceptual overlap, on which grounds schizotypy was pre-emptively excluded from the model, may well be more widespread and pernicious than could be successfully overcome by excluding only schizotypy while leaving all these other, equally if not even more conceptually overlapping, instruments still in the set of included predictors. The relative importances of those predictors remaining, when one omits the top ten most important predictors (of which ten only ‘aberrant salience’, ‘hallucinations’, ‘stress’ and ‘emotional regulation’ are not intrinsically constitutive of definitionally core and/or prototypical features proper to the target domain of persecutory delusions themselves) is not reported but may be of interest in this regard. Even more welcome an addition would be an analysis that excluded (as was done pre-emptively for schizotypy) all those factors that in a conceptual sense are inextricable from the phenomenology of the target domain itself. This is not to suggest that an important aetiological role for such conceptually-related predictors is unlikely to be genuine (indeed the causal models that prompted such scales’ inclusion are rooted in empirical and theoretical grounds for believing that they are), but the deck is unfairly stacked towards overemphasizing the importance aetiologically of those predictors that so happen to be measured by instruments that load directly on the same target construct whose aetiology is at issue. A full ranking of the importances of all predictors included in each analysis reported would allow the reader more novel information upon which to cultivate their own judgment as to the extent to which (for example) the conceptual overlap of ‘threat anticipation’ with the typical content of persecutory delusions themselves is driving most, or only a little, of its importance as a predictor of that outcome – based on the overall dominance of those predictors that share such an advantage, relative to those that do not and whose aetiological significance would be accordingly better supported (though not deductively necessitated) by a relatively high ranking given their lower degree of conceptual relatedness with the outcome being predicted. In this regard, it is especially interesting that (as the domain-general neurocomputational theory of delusion formation motivating their inclusion as potentially important causal factors thereof would predict they ought to be) the only two predictors of persecutory delusional ideas that both: (a) share a high predictive importance in predicting delusions regardless of thematic domain (i.e. regardless of whether the content of those delusions is persecutory (narrowly defined), persecutory and referential, or broadly defined to encompass items relating to many more of the common delusional themes as in the PDI-21); and (b) at the same time are nevertheless not broader predictors of high importance to general psychopathology...are the predictors ‘aberrant salience’ and ‘hallucinations’. The most influential theoretical framework implicating these experiences as a causally potent stimulus for inferential delusional formation (despite its ‘explanationist’ emphasis) would not posit such a clear division between ‘aetiological’ and ‘maintenance’ factors as is envisaged (for understandable pragmatic reasons) by the authors’ approach, but the authors’ conclusions that these factors especially stand out as potentially particularly important in aetiological terms (which are in their discussion acknowledged to not necessarily be as clearly distinct from ‘maintenance’ processes in reality as was by necessity assumed by methodological selection procedures) may well, on account of the foregoing considerations (and in light of them) be considered still yet more plausible and forceful.

Improving the Introduction and Explanation of a Novel ML Method:

As alluded to above, the methodology is likely to be novel to many, if not the vast majority, of researchers to whom this paper will be of direct interest. Though the authors can rely on computational methods being familiar to many (though probably not the majority) of these readers, the predictive 'black box' machine learning approach used here is of a quite different breed than the kinds of computational modelling approaches prevalent in research into the cognitive neuroscience of delusions. Nor will the universally high level of statistical literacy that should and can be assumed to characterise the audience for the paper at large render the methodology intelligible from the minimal descriptive treatment it is afforded here. This is a shame since an introduction to this novel methodology would likely be of great interest to many readers who will be encountering it here for the first time. Moreover, it is necessary to understand the methodology in much more detail than is provided or signposted by the paper, simply to be in a position to evaluate the logic of the paper's methods and the soundness of its conclusions. At present, the cited references do not constitute anything approaching a sufficient reading list to supply the relevant background (they are neither numerous enough, introductory enough, or specifically germane enough to the point being made by the sentence within which they are cited) – many terms that could usefully be explicated in the main text (e.g. 'nested cross-validation', 'inner folds/outer folds') are left undefined, and the average reader will have to be motivated to independently undertake a great deal of investigative reading elsewhere, if they are to be in a position to infer the reasons for each of a number of very cursorily brushed over methodological choices in the details of this machine learning approach's implementation. Though I am satisfied that the methodological choices are suitable for the purposes to which they are intended (as a result of my having undertaken to do this additional research in the service of writing this review) there is a missed opportunity to make the paper's conclusions more intelligible, which could be rectified were its novel machine learning methodology more engagingly and transparently presented. I believe this would greatly improve the paper's impact (in every sense of that word). The novelty of the application of random forest machine learning prediction methods to delusion research is a strength of the work - but only to the extent that the majority of its probable readership is able to meaningfully engage with and appreciate the logic underlying it.

Screening Procedure and Interpretability of High Scores Absent Concomitant Diagnostic Prevalences:

*The possibility of artificial inflation of persecutory belief questionnaire scores by subjects during the screening is thoughtfully considered, and this section of the discussion is convincing in its arguments. However, there is a striking disparity between the psychometric cut-off corresponding to the "likely" presence of clinical delusions of persecution (11) and the total absence of self-reported diagnosed psychotic illness within the 'moderately severe' group (scoring 11-18). Moreover, there were only two psychotic illness diagnoses among the equally large 'severe' group in the sample, - which is even more puzzling given that the lower threshold for inclusion in this latter group (a score of 18) has been previously shown to constitute a 'cut-off [that] is unlikely to identify incorrectly an individual as having a persecutory delusion when they do not' (Freeman D, Loe BS, Kingdon D, et al. The revised Green et al., Paranoid Thoughts Scale (R-GPTS): psychometric properties, severity ranges, and clinical cut-offs. *Psychological Medicine*. 2021;51(2):244-253. doi:10.1017/S0033291719003155).*

This puzzling disparity between the prevalence of psychotic disorders that would be expected in such high scorers, and the very low rates of same actually reported within these groups, warrants reflection. If the scale were functioning here (in the present study) in roughly the same

way as it did in the studies responsible for validating it and establishing its clinical cut-off points (i.e. if the scale in the present context were working simply 'as it should') then that would suggest a great number of participants within the upper two quadrants of severity can, with a high degree of confidence, be identified as individuals who very likely are suffering from undiagnosed and untreated clinical psychosis. If this is deemed the correct interpretation - was there any provision made in the ethics for contacting these individuals and signposting them to appropriate clinical attention/care?

Alternatively - if the very low rates of diagnosed psychosis self-reported by subjects are deemed concomitant with the true (low) underlying prevalence of that disorder, within the subset of the sample in question – then what does this imply about the mismatch between how the R-GPTS scale operates, and what it measures, in the present study's context - as compared to the contexts in which it was psychometrically validated and upon which the interpretation of it is grounded? One possibility is that other diagnoses (not captured by 'psychotic illnesses' as the term is used here, but themselves likely to be associated with delusional symptoms of these high levels of severity) were much more prevalent in the present sample than was psychotic illness itself – if the disparity between the rates of psychosis typical of such elevated R-GPTS scores, and the very low rates of diagnosed psychotic illness actually self-reported in these groups here, can be attributed to a high prevalence of such clinical diagnoses (e.g. borderline personality disorder), then perhaps this would be worth mentioning in the manuscript, as it would make sense of the otherwise-puzzling mismatch. This mismatch, left unreckoned with and unexplained, presently casts a rather tall shadow of doubt over the outcome measure's validity: suggesting that inflation of self-reported persecutory ideation in the screening phase was likely a particularly pernicious problem with the sampling strategy after all. A couple of sentences further consideration of the issue would do wonders to reassure the reader about the relative insignificance of this potential issue or (equally acceptably) to highlight it as an actual, substantive limitation.

31.07.2025

Dear Dr Pulcu,

Dear Reviewers,

We sincerely thank you for your positive feedback and for taking the time to review our manuscript. Your kind words and endorsement of our work are truly encouraging and greatly appreciated. It has been a pleasure to engage with your insightful and constructive comments throughout the revision process, and we are delighted that you found the revised manuscript to your satisfaction. We, too, look forward to seeing the paper published and accessible to the wider community.

Thank you again for your time and support!

With best wishes,

Saskia Denecke

(on behalf of all authors)

REVIEWERS' COMMENTS:

Reviewer #1 (Remarks to the Author):

The authors responded to all my queries. I am happy with the new manuscript. No more comments from my end. I look forward to see the paper available online.

Reviewer #2 (Remarks to the Author):

Dear authors,

Thank you very much for your reply, and for taking the time and care to implement such thoroughly satisfying revisions (to the manuscript/supplementaries) and thoughtful explanatory replies (in your rebuttal letter) as to fully address each of my comments, concerns and suggestions. Thank you also for your kind words of appreciation and helpful signposting! Since you have so admirably addressed all the points raised, all that remains on my end is to congratulate you on the new and improved version of your excellent paper, and to recommend it unreservedly to the editors for publication in its current form. Reviewing your work has been a pleasure, and I am sure that your contribution will be widely read and enjoyed, its very innovative and impressive stuff indeed. Congratulations and best wishes.